# Personalized Federated Learning with Communication Compression

**El Houcine Bergou**[*]                                                     *Elhoucine.bergou@um6p.ma*
*College of Computing*
*Mohammed VI Polytechnic University*
*Ben Guerir, Morocco*

**Konstantin Burlachenko**                                    *konstantin.burlachenko@kaust.edu.sa*
*King Abdullah University of Science and Technology, KSA*

**Aritra Dutta**[†]                                                              *aritra.dutta@ucf.edu*
*Artificial Intelligence Initiative*
*University of Central Florida*
*Orlando, Florida-32816*

**Peter Richtárik**                                                    *peter.richtarik@kaust.edu.sa*
*King Abdullah University of Science and Technology, KSA*

**Reviewed on OpenReview:** *https://openreview.net/forum?id=dZugyhbNFY*

## Abstract

In contrast to training traditional machine learning (ML) models in data centers, federated learning (FL) trains ML models over local datasets on resource-constrained heterogeneous edge devices. Existing FL algorithms aim to learn a single global model for all participating devices, which may not be helpful to all devices participating in the training due to the heterogeneity of the data across the devices. Recently, Hanzely and Richtárik (2020) proposed a new formulation for training personalized FL models aimed at balancing the trade-off between the traditional global model and the local models that could be trained by individual devices using their private data only. They derived a new algorithm, called *loopless gradient descent* (L2GD), to solve it and showed that this algorithm leads to improved communication complexity guarantees in regimes when more personalization is required. In this paper, we equip their L2GD algorithm with a *bidirectional* compression mechanism to further reduce the communication bottleneck between the local devices and the server. Unlike other compression-based algorithms used in the FL setting, our compressed L2GD algorithm operates on a probabilistic communication protocol, where communication does not happen on a fixed schedule. Moreover, our compressed L2GD algorithm maintains a similar convergence rate as vanilla SGD without compression. To empirically validate the efficiency of our algorithm, we perform diverse numerical experiments on both convex and non-convex problems and use various compression techniques.[1]

## 1 Introduction

We live in the era of big data, and edge devices have become a part of our daily lives. While the training of ML models using the diverse data stored on these devices is becoming increasingly popular, the traditional data center-based approach to training them faces serious *privacy issues* and has to deal with *high communication*

---

[*]Equal contribution

[†]Equal contribution

[1]Our repository is available online: `https://github.com/burlachenkok/compressed-fl-l2gd-code`.

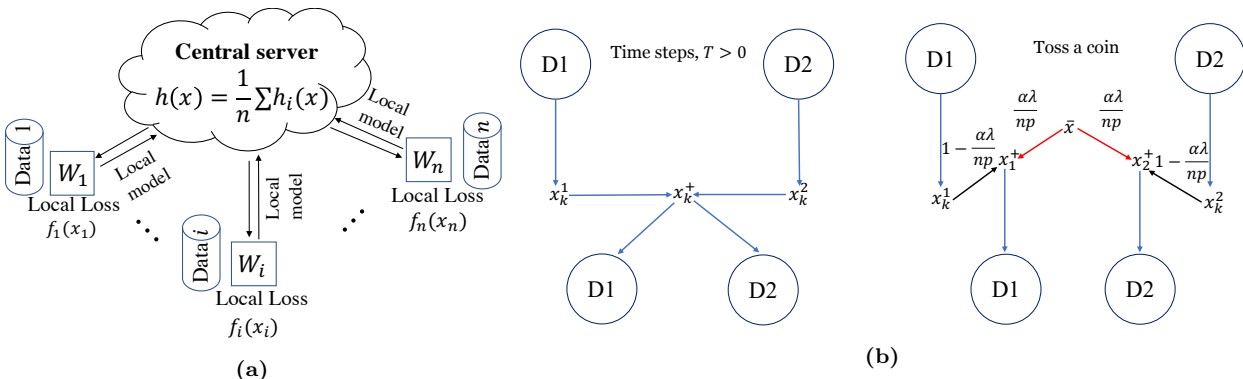

Figure 1: **(a)Training $n$ local devices, $\{W_i\}$ on the loss, $f_i$ of their local model, $x_i$ with a central server/master node, where $h_i$ penalizes for dissimilarity between the local model, $x_i$ and the average of all local models, $\bar{x}$. (b)FedAvg (McMahan et al., 2017) and L2GD (Hanzely & Richtárik, 2020) algorithm on 2 devices (D1 and D2). Unlike FedAvg, L2GD does not communicate after a fixed $T$ local steps, it communicates based on a probabilistic protocol.**

*and energy cost* associated with the transfer of data from users to the data center (Dean et al., 2012). *Federated learning* (FL) provides an attractive alternative to the traditional approach as it aims to train the models directly on *resource constrained* heterogeneous devices without any need for the data to leave them (Konečný et al., 2016; Kairouz et al., 2019).

The prevalent paradigm for training FL models is empirical risk minimization—to train a *single global model* using the aggregate of all the training data stored across all participating devices. Among the popular algorithms for training FL models for this formulation belong FedAvg (McMahan et al., 2017), Local GD (Khaled et al., 2019; 2020), local SGD (Stich, 2019; Khaled et al., 2020; Gorbunov et al., 2021) and Shifted Local SVRG (Gorbunov et al., 2021). All these methods require the participating devices to perform a local training procedure (e.g., by taking multiple steps of some optimization algorithm) and subsequently communicate the resulting model to an orchestrating server for aggregation; see Figure 1a. This process is repeated until a model of suitable qualities is found. For more variants of local methods and further pointers to the literature, we refer the reader to Gorbunov et al. (2021).

## 1.1 Personalized FL

In contrast, Hanzely & Richtárik (2020) recently introduced a new formulation of FL as an alternative to the existing "single-model-suits-all" approach embodied by empirical risk minimization. Their formulation explicitly aims to find a *personalized* model for every device; see Figure 1a. In particular, Hanzely & Richtárik (2020) considered the formulation[2]

$$\min_{x \in \mathbb{R}^{nd}} [F(x) := f(x) + h(x)] \tag{1}$$

for simultaneous training of $n$ personalized FL models $x_1, \ldots, x_n \in \mathbb{R}^d$ for $n$ participating devices. They chose

$$f(x) := \tfrac{1}{n} \sum_{i=1}^{n} f_i(x_i), \quad \text{and} \quad h(x) := \tfrac{1}{n} \sum_{i=1}^{n} h_i(x_i),$$

where $f_i$ represents the loss of model $x_i$ over the local data stored on device $i$. Function $h_i$ penalizes for dissimilarity between the local model $x_i$ and the average of all local models $\bar{x} := \tfrac{1}{n} \sum_{i=1}^{n} x_i$, and is defined to be

$$h_i(x) = \tfrac{\lambda}{2} \|x_i - \bar{x}\|_2^2,$$

where $\lambda > 0$ controls for the strength of this penalization. At one extreme, $\lambda \to \infty$ forces the local models to be equal to their average, and hence, mutually identical. Therefore, equation 1 reduces to the classical

---

[2]Zhang et al. (2015) considered a similar model in a different context and with different motivations.

empirical risk minimization formulation of FL

$$\min_{z \in \mathbb{R}^d} \tfrac{1}{n} \sum_{i=1}^{n} f_i(z).$$

On the other hand, for $\lambda = 0$ problem equation 1 is equivalent to each client (node) training independently using their data only. In particular, the $i^{\text{th}}$ client solves

$$\min_{x_i \in \mathbb{R}^d} f_i(x_i).$$

By choosing $\lambda$ to a value in between these two extremes, i.e., $0 < \lambda < \infty$, we control for the level of similarity we want the personalized models $\{x_i\}_{i=1}^{n}$ to possess.

We remark that local methods such as FedAvg by McMahan et al. (2017) (also see similar methods in (Haddadpour et al., 2019b; Stich, 2019; Wang & Joshi, 2019; Zhou & Cong, 2018; Lin et al., 2020)), are popular for training FL models. Nevertheless, their main drawback in the heterogeneous setting with data and device heterogeneity is inefficient communication. Hanzely & Richtárik (2020) proposed this new personalization to tackle heterogeneous data, and we are using their model to build our compressed, personalized FL.

To solve equation 1, Hanzely & Richtárik (2020) proposed a *probabilistic* gradient descent algorithm for which they coined the name loopless local gradient descent (L2GD). Hanzely & Richtárik (2020) shows how L2GD can be interpreted as a simple variant of FedAvg, typically presented as a method for solving the standard empirical risk minimization (ERM) formulation of FL. However, alongside Hanzely & Richtárik (2020) argue, L2GD is better seen as an algorithm for solving the personalized FL formulation equation 1. By doing so, they interpret the nature of local steps in classical FL: the role of local steps in classical FL methods is to provide personalization and not communication efficiency as was widely believed—FedAvg can diverge on highly non-identical data partitions (McMahan et al., 2017). Instead, communication efficiency in local methods comes from their tendency to gear towards personalization, and personalized models are provably easier to train.

**Communication compression.** We observe that the *L2GD algorithm does not support any compression mechanism* for the master-worker and worker-master communication that needs to happen—This is the starting point of our work. *We believe that equipping personalized FL with fast and theoretically tractable communication compression mechanisms is an important open problem.*

In distributed training of deep neural network (DNN) models, synchronous data-parallelism (Dean et al., 2012) is most widely used and adopted by mainstream deep learning toolkits (such as `PyTorch, TensorFlow`). However, exchanging the stochastic gradients in the network for aggregation creates a communication bottleneck, and this results in slower training (Xu et al., 2021a). One way to save on communication costs is to use compression operators (Alistarh et al., 2017; Horváth et al., 2019; Xu et al., 2021a). Gradient compression techniques, such as quantization (Alistarh et al., 2017; Bernstein et al., 2018; Horváth et al., 2019; Beznosikov et al., 2020; Safaryan et al., 2020), sparsification (Suresh et al., 2017; Konečný & Richtárik, 2018; Aji & Heafield, 2017; Sahu et al., 2021; Stich et al., 2018; Dutta et al., 2020; Safaryan et al., 2020), hybrid compressors (Strom, 2015; Basu et al., 2019), and low-rank methods (Vogels et al., 2019) have been proposed to overcome this issue. [3]

Although recent works have introduced compression in traditional FL formulation (Konečný et al., 2016; Reisizadeh et al., 2020; Philippenko & Dieuleveut, 2020; Shlezinger et al., 2020; Amiri et al., 2020; Xu et al., 2021b); except (Horváth et al., 2019; Philippenko & Dieuleveut, 2020; Gorbunov et al., 2020; Amiri et al., 2020), others use compression only for the *throughput limited uplink* channel, that is, to upload the local models from the devices to the central server. But limited bandwidth in the downlink channel may pose a communication latency between the server and the devices and consequently, slow down the training; see detailed discussion in §2. As of now, no study combines *bidirectional* compression techniques with a probabilistic communication protocol in the FL set-up by using a mixture of a local and global model as in equation 1. In this work, we combine these aspects and make subsequent contributions.

---

[3]Model compression (Guo, 2018; Chraibi et al., 2019) is orthogonal to gradient compression and not in the scope of this work.

### 1.2 Contributions

(*i*) **L2GD algorithm with bidirectional compression.** Communication compression is prevalent in recent local FL training algorithms, but these algorithms are not robust to data and device heterogeneity. L2GD by Hanzely & Richtárik (2020) remedies this issue by introducing personalization in FL. However, integrating compression with the L2GD algorithm is a nontrivial task—unlike other FL algorithms, L2GD does not communicate after fixed local steps, it communicates based on a probabilistic protocol; see §3 and Figure 1b. Additionally, due to this probabilistic protocol, the communication involves local model updates, as well as gradients; see §3. To reduce the communication bottleneck in L2GD, we use compression techniques on top of its probabilistic communication at both master and the participating local devices; see §4. To the best of our knowledge, we are the first to integrate *bidirectional compression techniques* with a probabilistic communication in the FL set-up, and we call our algorithm *compressed L2GD*; see Algorithm 1.

(*ii*) **Convergence analysis.** In §5, we prove the convergence of our *compressed L2GD* algorithm based on the most recent theoretical development, such as expected smoothness as in Gower et al. (2019). Admittedly, convergence analysis of first-order optimization algorithms with bidirectional compression exists in the literature, see (Tang et al., 2019; Horváth et al., 2019; Amiri et al., 2020; Dutta et al., 2020), integrating arbitrary unbiased compressors with a probabilistic communication protocol into personalized FL, and showing convergence are nontrivial and the first one in its class. Our compressed L2GD algorithm maintains a similar asymptotic convergence rate as the baseline vanilla SGD without compression in both strongly convex and smooth nonconvex cases; see Theorem 1 and 2 in §5.

(*iii*) **Optimal rate and communication.** We optimized the complexity bounds of our algorithm as a function of the parameters involved in the algorithm. This leads us to the *optimal* setting of our algorithm. Mainly, we derived the optimal expected number of local steps to get the optimal iteration complexity and communication rounds; see §6. Although our analysis is based on some hard-to-compute constants in real life, e.g., the Lipchitz constant, this may help the practitioners to get an insight into the iteration complexity and communication trade-off; see Theorem 3 and 4 in §6.

(*iv*) **Empirical study.** We perform diverse numerical experiments on synthetic and real datasets by using both convex and non-convex problems (using 4 DNN models) and invoking various compression techniques; see details in §7, Table 1. In training larger DNN models, to obtain the same global Top-1 test accuracy, compressed L2GD reduces the communicated data volume (bits normalized by the number of local devices or clients, #bits/n), from $10^{15}$ to $10^{11}$, rendering approximately $10^4$ times improvement compared to FedAvg; see §7.2. Moreover, L2GD with natural compressor (that by design has smaller variance) empirically behaves the best and converges approximately 5 times faster, and reaches the best accuracy on both train and the test sets compared to no-compression FedOpt (Reddi et al., 2020) baseline; see §7.2 and §A.2. These experiments validate the effect of the parameters used and, the effect of compressors, and show the efficiency of our algorithm in practice.

## 2 Related Work

Numerous studies are proposed to reduce communication but not all of them are in the FL setting. In this scope, for completeness, we quote a few representatives from each class of communication-efficient SGD.

Smith et al. (2017) proposed a communication-efficient primal-dual optimization that learns separate but related models for each participating device. FedAvg by McMahan et al. (2017) performs local steps on a subset of participating devices in an FL setting. Similar to FedAvg, but without data and device heterogeneity, (Haddadpour et al., 2019b; Stich, 2019; Wang & Joshi, 2019; Zhou & Cong, 2018; Lin et al., 2020) independently proposed local SGD, where several local steps are taken on the participating devices before periodic communication and averaging the local models. While FedProx by Li et al. (2020) is a generalization of FedAvg, SCAFFOLD uses a variance reduction to correct local updates occurring from non-i.i.d data in FedAvg. From the system's perspective, on `TensorFlow`, Bonawitz et al. (2019) built a FL system on mobile devices.

Compression has also been introduced in the FL setup. Shlezinger et al. (2020) combined universal vector quantization with FL for throughput limited uplink channel. In FedPAQ by Reisizadeh et al. (2020), each local device sends a compressed difference between its input and output model to the central server, after computing the local updates for a fixed number of iterations. While Amiri et al. (2020) used a bidirectional compression in FL set-up, Philippenko & Dieuleveut (2020) combined it with a memory mechanism or error feedback (Stich et al., 2018). For a unified analysis of compression in FL we refer to (Haddadpour et al., 2021).

Among other proposed communication-efficient SGDs, parallel restarted SGD (Yu et al., 2019a) reduces the number of communication rounds compared to the baseline SGD. Haddadpour et al. (2019a) showed that redundancy reduces residual error as compared with the baseline SGD where all nodes can sample from the complete data and this leads to lower communication overheads. CoCoA by (Jaggi et al., 2014), and Dane by Shamir et al. (2014) perform several local steps and hence fewer communication rounds before communicating with the other workers. Lazily aggregated gradient (LAG) algorithm by Chen et al. (2018a) selects a subgroup of workers and uses their gradients, instead of obtaining a fresh gradient from each worker in each iteration. For communication-efficient local SGD see (Gao et al., 2021).

In decentralized training, where the nodes only communicate with their neighbors, Koloskova et al. (2019) implemented an *average consensus* where the nodes can communicate to their neighbors via a fixed communication graph. Li et al. (2018) proposed Pipe-SGD—a framework with decentralized pipelined training and balanced communication.

Personalization in FL is a growing research area. Arivazhagan et al. (2019) proposed FedPer to mitigate statistical heterogeneity of data; also, see adaptive personalized FL algorithm in (Deng et al., 2020). Mei et al. (2021) proposed to obtain personalization in FL by using layer-wise parameters, and two-stage training; also, see (Ma et al., 2022) and model personalization in (Shen et al., 2022). Shamsian et al. (2021) trained a central hypernetwork model to generate a set of personalized models for the local devices. Li et al. (2021) proposed Hermes—a communication-efficient personalized FL, where each local device identifies a small subnetwork by applying the structured pruning, communicates these subnetworks to the server and the devices, the server performs the aggregation on only overlapped parameters across each subnetwork; also, see Pillutla et al. (2022) for partial model personalization in FL. DispFL is another communication-efficient personalized FL algorithm proposed by Dai et al. (2022). In recent work, Zhang et al. (2021) introduces personalization by calculating optimal weighted model combinations for each client without assuming any data distribution. For a connection between personalization in FL and model-agnostic-meta-learning (MAML), see (Fallah et al., 2020). Additionally, we refer to the surveys (Kulkarni et al., 2020; Tan et al., 2021) for an overview of personalization in FL.

## 3 Background and Preliminaries

**Notation.** For a given vector, $x \in \mathbb{R}^{nd}$, by $x_i$ we denote the $i^{\text{th}}$ subvector of $x$, and write $x = \left(x_1^\top, \ldots, x_n^\top\right)^\top$, where $x_i \in \mathbb{R}^d$. We denote the $i^{\text{th}}$ component of $x$ by $x_{(i)}$ and $\|x\|$ represents its Euclidean norm. By $[n]$ we denote the set of indexes, $\{1, \ldots, n\}$. By $\mathbb{E}_\xi(\cdot)$ we define the expectation over the randomness of $\xi$ conditional to all the other potential random variables. The operator, $\mathcal{C}(\cdot) := \left(\mathcal{C}_1(\cdot)^\top, \ldots, \mathcal{C}_n(\cdot)^\top\right)^\top : \mathbb{R}^{nd} \to \mathbb{R}^{nd}$ denotes a compression operator with each $\mathcal{C}_i(\cdot)$ being compatible with the size of $x_i$. Denote $Q := [I, I, \ldots, I]^\top \in \mathbb{R}^{nd \times d}$, where $I$ denotes the identity matrix of $\mathbb{R}^{d \times d}$. With our Assumptions that we will introduce later in the paper, the problem in equation 1 has a unique solution, which we denote by $x^*$ and we define $\bar{x}^*$ as $\bar{x}^* = \frac{1}{n} \sum_{i=1}^n x_i^*$. By $|S|$ we denote the cardinality of a set, $S$.

**Loopless local gradient descent (L2GD).** We give a brief overview of the loopless local gradient descent (L2GD) algorithm by Hanzely & Richtárik (2020) to solve equation 1 as a two sum problem. At each iteration, to estimate the gradient of $F$, L2GD samples either the gradient of $f$ or the gradient of $h$ and updates the local models via:

$$x_i^{k+1} = x_i^k - \alpha G_i(x^k), \ i = 1, \ldots, n,$$

where $G_i(x^k)$ for $i = 1, \ldots, n$, is the $i^{\text{th}}$ block of the vector

$$G(x^k) = \begin{cases} \frac{\nabla f(x^k)}{1-p} & \text{with probability } 1-p, \\ & \textbf{(Local gradient step)} \\ \frac{\nabla h(x^k)}{p} & \text{with probability } p, \\ & \textbf{(Aggregation step)} \end{cases}$$

where $0 < p < 1$, $\nabla f(x^k)$ is the gradient of $f$ at $x^k$, and $\nabla_i h(x^k) = \frac{\lambda}{n}\left(x_i^k - \bar{x}^k\right)$ is the $i^{\text{th}}$ block of the gradient of $h$ at $x^k$.

In this approach, there is a hidden communication between the local devices because in aggregation steps they need the average of the local models. That is, the communication occurs when the devices switch from a local gradient step to an aggregation step. Note that there is no need for communication between the local devices when they switch from an aggregation step to a local gradient step. There is also no need for communication after two consecutive aggregation steps since the average of the local models does not change in this case. If $k$ and $k+1$ are both aggregation steps, we have $\bar{x}^{k+1} = \frac{1}{n}\sum_{i=1}^{n} x_i^{k+1} = \frac{1}{n}\sum_{i=1}^{n} x_i^k - \frac{\alpha\lambda}{n}\frac{1}{n}\sum_{i=1}^{n}\left(x_i^k - \bar{x}^k\right) = \bar{x}^k$.

## 4 Compressed L2GD

Now, we are all set to describe the compressed L2GD algorithm for solving (1). We start by defining how the compression operates in our set-up.

### 4.1 Compressed communication

Recall that the original L2GD algorithm has a probabilistic communication protocol—the devices do not communicate after every fixed number of local steps. The communication occurs when the devices switch from a local gradient step to an aggregation. Therefore, instead of using the compressors in a fixed time stamp (after every $T > 0$ iterations, say), each device $i$ requires to compress its local model $x_i$ when it needs to communicate it to the master, based on the probabilistic protocol. We assume that device $i$ uses the compression operator, $\mathcal{C}_i(\cdot) : \mathbb{R}^d \to \mathbb{R}^d$. Moreover, another compression happens when the master needs to communicate with the devices. We assume that the master uses the compression operator, $\mathcal{C}_M(\cdot) : \mathbb{R}^d \to \mathbb{R}^d$. Therefore, the compression is used in uplink and downlink channels similar to Dutta et al. (2020); Horváth et al. (2019), but occurs in a probabilistic fashion. There exists another subtlety—although the model parameters (either from the local devices or the global aggregated model) are communicated in the network for training the FL model via compressed L2GD, the compressors that we use in this work are the compressors used for gradient compression in distributed DNN training; see (Xu et al., 2021a).

### 4.2 The algorithm

Note that, in each iteration $k \geq 0$, there exists a random variable, $\xi_k \in \{0, 1\}$ with $P(\xi_k = 1) = p$ and $P(\xi_k = 0) = 1 - p$. If $\xi_k = 0$, all local devices at iteration $k$ perform one local gradient step. Otherwise (if $\xi_k = 1$), all local devices perform an aggregation step. However, to perform an aggregation step, the local devices need to know the average of the local models. If the previous iteration (i.e., $k - 1^{\text{th}}$ iteration) was an aggregation step (i.e., $\xi_{k-1} = 1$) then at the current iteration the local devices can use the same average as the one at iteration $k - 1$ (recall, the average of the local models does not change after two consecutive aggregation steps). Otherwise, a communication happens with the master to compute the average. In this case, each local device $i$ compresses its local model $x_i^k$ to $\mathcal{C}_i(x_i^k)$ and communicates the result to the master. The master computes the average based on the compressed values of local models:

$$\bar{y}^k := \frac{1}{n}\sum_{j=1}^{n} \mathcal{C}_j(x_j^k),$$

then it compresses $\bar{y}^k$ to $\mathcal{C}_M(\bar{y}^k)$ by using a compression operator at the master's end and communicates it back to the local devices. The local devices further perform an aggregation step by using $\mathcal{C}_M(\bar{y}^k)$ instead of

the exact average. This process continues until convergence. From Algorithm 1, we have, for $i = 1, \dots, n$:

$$x_i^{k+1} = x_i^k - \eta G_i(x^k),$$

where

$$G_i(x^k) = \begin{cases} \frac{\nabla f_i(x_i^k)}{n(1-p)} & \text{if } \xi_k = 0 \\ \frac{\lambda}{np}\left(x_i^k - \mathcal{C}_M(\bar{y}^k)\right) & \text{if } \xi_k = 1 \ \& \ \xi_{k-1} = 0, \\ \frac{\lambda}{np}\left(x_i^k - \bar{x}^k\right) & \text{if } \xi_k = 1 \ \& \ \xi_{k-1} = 1. \end{cases}$$

We give the pseudo code in Algorithm 1.

---

**Algorithm 1** Compressed L2GD

---

**Input:** $\{x_i^0\}_{i=1,\dots,n}$, stepsize $\eta > 0$, probability $p$, $\xi_{-1} = 1$, $\bar{x}^{-1} = \frac{1}{n}\sum_{i=1}^n x_i^0$.
**for** $k = 0, 1, 2, \dots$ **do**
  **Draw:** $\xi_k = 1$ with probability $p$
  **if** $\xi_k = 0$ **then**
    **on all devices:** $x_i^{k+1} = x_i^k - \frac{\eta}{n(1-p)}\nabla f_i(x_i^k)$ for $i \in [n]$
  **else**
    **if** $\xi_{k-1} = 0$ **then**
      **on all devices:** Compress $x_i^k$ to $\mathcal{C}_i(x_i^k)$ and communicate $\mathcal{C}_i(x_i^k)$ to the master
      **on master:** receive $\mathcal{C}_i(x_i^k)$ from the device $i$, for all $i \in [n]$ compute $\bar{y}^k := \frac{1}{n}\sum_{j=1}^n \mathcal{C}_j(x_j^k)$
      compress $\bar{y}^k$ to $\mathcal{C}_M(\bar{y}^k)$ and communicate it to all devices
      **on all devices:** Perform aggregation step $x_i^{k+1} = x_i^k - \frac{\eta\lambda}{np}\left(x_i^k - \mathcal{C}_M(\bar{y}^k)\right)$
    **else**
      **on all devices:** $\bar{x}^k = \bar{x}^{k-1}$, Perform aggregation step $x_i^{k+1} = x_i^k - \frac{\eta\lambda}{np}\left(x_i^k - \bar{x}^k\right)$
    **end if**
  **end if**
**end for**

---

**Remark 1** *The initialization, $\xi_{-1}$ is not important, as it does not impact the Algorithm. In Algorithm 1, we start the for loop (for $k = 0, 1, \dots$) by drawing $\xi_k$. At iteration $k$, we perform an aggregation step if $\xi_k = 1$. That is why to be consistent in the algorithm, we initialized $\xi$ at iteration "-1" by 1 and initialized $\bar{x}_{-1}$ by the average of the initial models, $x_i^0$.*

**Remark 2** *All local devices have access to the same value of $\xi$. We assumed that $\xi$ is drawn at random at the server side and broadcasted to the local devices.*

## 5 Convergence Analysis

With the above setup, we now prove the convergence of Algorithm 1; see detailed proofs in §A.1.

### 5.1 Assumptions

We make the following general assumptions in this paper.

**Assumption 1** *For $i = 1, \dots, n$:*

- *The compression operator, $\mathcal{C}_i(\cdot) : \mathbb{R}^d \to \mathbb{R}^d$ is unbiased,*

$$\mathbb{E}_{\mathcal{C}_i}[\mathcal{C}_i(x)] = x, \quad \forall x \in \mathbb{R}^d.$$

- *There exists constant, $\omega_i > 0$ such that the variance of $\mathcal{C}_i$ is bounded as follows:*

$$\mathbb{E}_{\mathcal{C}_i}\left[\|\mathcal{C}_i(x) - x\|^2\right] \le \omega_i\|x\|^2, \forall x \in \mathbb{R}^d.$$

- *The operators, $\{\mathcal{C}_i(\cdot)\}_{i=1}^n$ are independent from each other, and independent from $\xi_k$, for all $k \geq 0$.*

- *The compression operator, $\mathcal{C}_M(\cdot)$ is unbiased, independent from $\{\mathcal{C}_i\}_{i=1}^n$ and has compression factor, $\omega_M$.*

From the above assumption we conclude that for all $x \in \mathbb{R}^d$, we have

$$\mathbb{E}_{\mathcal{C}_i}\left[\|\mathcal{C}_i(x)\|^2\right] \leq (1+\omega_i)\|x\|^2.$$

The following lemma characterizes the compression factor, $\omega$ of the joint compression operator, $\mathcal{C}(\cdot) = \left(\mathcal{C}_1(\cdot)^\top, \ldots, \mathcal{C}_n(\cdot)^\top\right)^\top$ as a function of $\omega_1, \ldots, \omega_n$.

**Lemma 1** *Let $x \in R^{nd}$, then*

$$\mathbb{E}_{\mathcal{C}}\left[\|\mathcal{C}(x)\|^2\right] \leq (1+\omega)\|x\|^2,$$

*where $\omega = \max_{i=1,\ldots,n}\{\omega_i\}$.*

For the convergence of strongly convex functions, we require an additional assumption on the function $f$ as follows.

**Assumption 2** *We assume that $f$ is $L_f$-smooth and $\mu$-strongly convex.*

## 5.2 Auxiliary results

Before we state our main convergence theorem, we state several intermediate results needed for the convergence. In the following two lemmas, we show that based on the randomness of the compression operators, in expectation, we recover the exact average of the local models and the exact gradients for all iterations.

**Lemma 2** *Let Assumption 1 hold, then for all $k \geq 0$, $\mathbb{E}_{\mathcal{C},\mathcal{C}_M}\left[\mathcal{C}_M(\bar{y}^k)\right] = \bar{x}^k$.*

**Lemma 3** *Let Assumptions 1 hold. Then for all $k \geq 0$, knowing $x^k$, $G(x^k)$ is an unbiased estimator of the gradient of function $F$ at $x^k$.*

Our next lemma gives an upper bound on the iterate at each iteration. This bound is composed of two terms—the optimality gap, $F(x^k) - F(x^*)$, and the norm at the optimal point, $x^*$.

**Lemma 4** *Let Assumption 2 hold, then*

$$\left\|x^k\right\|^2 \leq \frac{4}{\mu}\left(F(x^k) - F(x^*)\right) + 2\left\|x^*\right\|^2.$$

Lemma 5 helps us to prove the expected smoothness property (Gower et al., 2019). The bound in Lemma 5 is composed of—the optimality gap, the difference between the gradients of $h$ at $x^k$ and $x^*$, and an extra constant, $\beta$ that depends on the used compressors.

**Lemma 5** *Let Assumptions 1 and 2 hold. Then*

$$\mathcal{A} := \mathbb{E}_{\mathcal{C}_M,\mathcal{C}}\left\|x^k - Q\mathcal{C}_M(\bar{y}^k) - x^* + Q\mathcal{C}_M(\bar{y}^*)\right\|^2 \leq \frac{4n^2}{\lambda^2}\left\|\nabla h(x^k) - \nabla h(x^*)\right\|^2 + \alpha\left(F(x^k) - F(x^*)\right) + \beta,$$

*where $\bar{y}^* := \frac{1}{n}\sum_{j=1}^n \mathcal{C}_j(x_j^*)$, $\alpha := \frac{4(4\omega + 4\omega_M(1+\omega))}{\mu}$, and $\beta := 2\left(4\omega + 4\omega_M(1+\omega)\right)\left\|x^*\right\|^2 + 4\mathbb{E}_{\mathcal{C}_M,\mathcal{C}}\left\|Q\mathcal{C}_M(\bar{y}^*) - Q\bar{x}^*\right\|^2.$*

Lemma 6 is the final result that (together with Lemma 5) shows the expected smoothness property and gives an upper bound on the stochastic gradient. This bound is composed of three terms—the optimality gap, $F(x^k) - F(x^*)$, the expected norm of the stochastic gradient at the optimal point, $\mathbb{E}\|G(x^*)\|^2$, and some other quantity that involves interplay between the parameters used in Algorithm 1 and the used compressor.

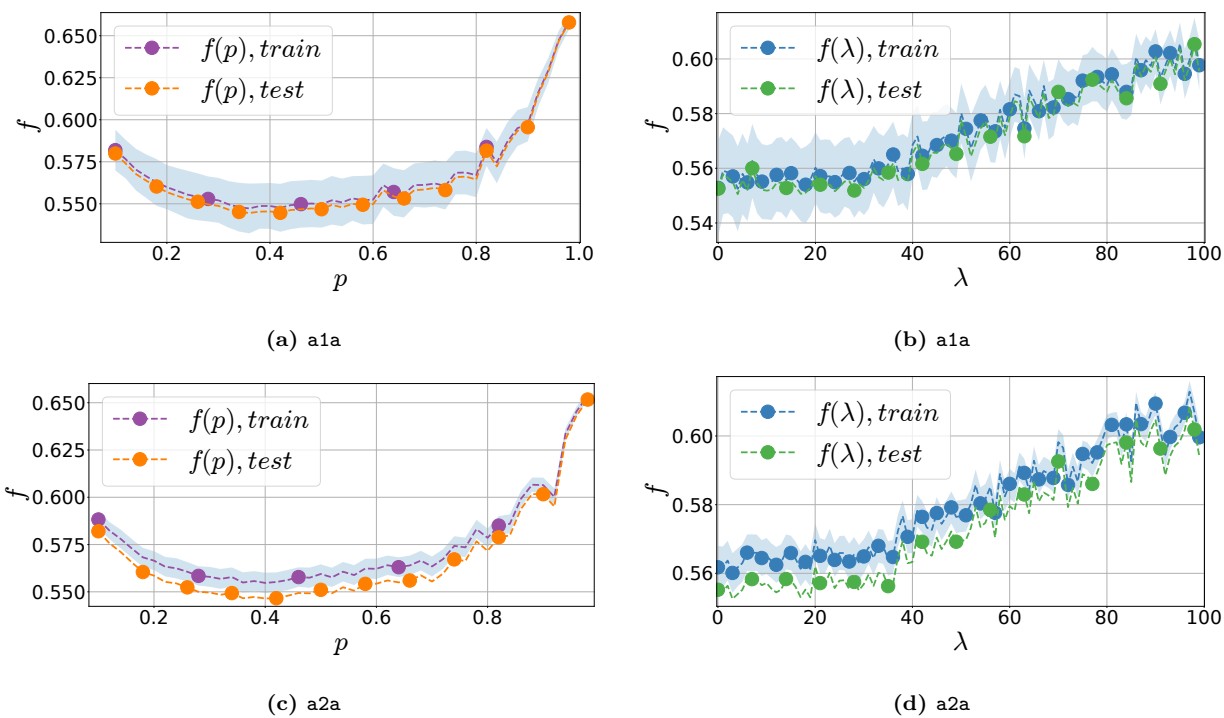

**Figure 2:** **Uncompressed L2GD on $n = 5$ workers for $f_i(x)$ to be local empirical risk minimization for logistic regression loss with $\ell_2$ regularization for local data $D_i$.** We show the loss, $f$ as a function of $p$ and $\lambda$ obtained after $K = 100$ iterations of Algorithm 1 with $\mathcal{C}$ an identity compressor (no compression). **(a) a1a dataset, $d = 124, \lambda = 10$, (b) a1a dataset, $d = 124, p = 0.65$, (c) a2a dataset, $d = 124, \lambda = 10$, (d) a2a dataset, $d = 124, p = 0.65$.**

**Lemma 6 (Expected Smoothness)** *Let Assumptions 1 and 2 hold, then*

$$\mathbb{E}\left[\|G(x^k)\|^2 | x^k\right] \leq 4\gamma\left(F(x^k) - F(x^*)\right) + \delta, \tag{2}$$

*where*

$$\gamma := \frac{\alpha\lambda^2(1-p)}{2n^2p} + \max\left\{\frac{L_f}{(1-p)}, \frac{\lambda}{n}\left(1 + \frac{4(1-p)}{p}\right)\right\}$$

*and*

$$\delta := \frac{2\beta\lambda^2(1-p)}{n^2p} + 2\mathbb{E}\|G(x^*)\|^2.$$

**Remark 3** *If there is no compression, the operators, $\mathcal{C}_i(\cdot)$, for $i \in [n]$, and $\mathcal{C}_M(\cdot)$ are equal to identity. The compression constants, $\omega_i$, for $i \in [n]$, and $\omega_M$ are equal to zero. Therefore, $\alpha = \beta = 0$, and the factor 4 in the formula of $\gamma$ can be replaced by 1 and thus*

$$\delta = \frac{2\beta\lambda^2(1-p)}{n^2p} + 2\mathbb{E}\|G(x^*)\|^2 = 2\mathbb{E}\|G(x^*)\|^2,$$

*with*

$$\gamma = \max\left\{\frac{L_f}{(1-p)}, \frac{\lambda}{n}\left(1 + \frac{(1-p)}{p}\right)\right\} = \max\left\{\frac{L}{n(1-p)}, \frac{\lambda}{np}\right\},$$

*where $L = nL_f$. Same constants arise in the expected smoothness property in Hanzely & Richtárik (2020).*

For nonconvex convergence of our algorithm, we cannot use the expected smoothness of Lemma 6, as it requires $\mu$-strong convexity of the loss function. Therefore, similar to (Sahu et al., 2021; Stich & Karimireddy, 2020) we require a new assumption to bound the stochastic gradient, $G(x^k)$. Let $G(x^k)$ is of the form: $G(x^k) = \nabla F(x^k) + \zeta_k$, where $\zeta_k$ is the stochastic noise on the gradient.

**Assumption 3** *There exist $M, \sigma^2 \geq 0$, such that $\mathbb{E}[\|G(x^k)\|^2 \mid x^k] \leq M\|\nabla F(x^k)\|^2 + \sigma^2$.*

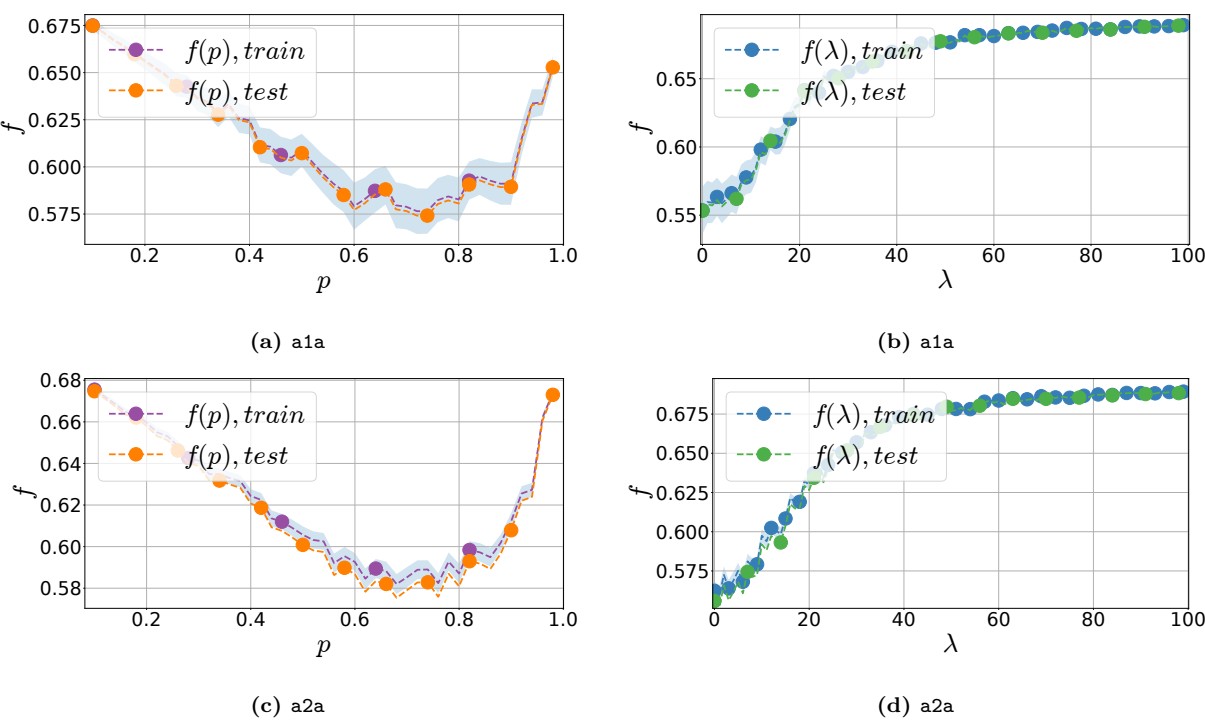

**Figure 3: Compressed L2GD on** $n = 5$ **workers for** $f_i(x)$ **to be local empirical risk minimization for logistic regression loss with** $\ell_2$ **regularization for local data** $D_i$**. We show the loss,** $f$ **as a function of** $p$ **and** $\lambda$ **obtained after** $K = 100$ **iterations of Algorithm 1 with with natural compressor at the workers and identity compressor (no compression) at the master. (a)** a1a **dataset,** $d = 124, \lambda = 10$**, (b)** a1a **dataset,** $d = 124, p = 0.65$**, (c)** a2a **dataset,** $d = 124, \lambda = 10$**, (d)** a2a **dataset,** $d = 124, p = 0.65$**.**

## 5.3 Main result

We now state the convergence result for Algorithm 1 for both strongly convex and nonconvex cases.

**Theorem 1** *(Strongly convex case) Let Assumptions 1 and 2 hold. If $\eta \leq \frac{1}{2\gamma}$, then*

$$\mathbb{E}\left\|x^k - x^*\right\|^2 \leq \left(1 - \frac{\eta\mu}{n}\right)^k \left\|x^0 - x^*\right\|^2 + \frac{n\eta\delta}{\mu}.$$

**Proof 1** *The proof follows directly from Lemma 3, 6, and Theorem 3.1 from Gower et al. (2019).*

**Theorem 2** *(Non convex case) Let Assumptions 1 and 3 hold. Assume also that $F$ is $L_f$-smooth, bounded from below by $F(x^*)$. Then to reach a precision, $\epsilon > 0$, set the stepsize, $\eta = \min\{\frac{1}{L_f M}, \frac{\epsilon^2}{2L_f\sigma^2}\}$, such that for $K \geq \frac{4L_f M(F(x^0) - F(x^*))}{\epsilon^2}$, we have $\min_{k=0,1,...,K} \mathbb{E}\|\nabla F(x^k)\|_2 \leq \epsilon$.*

**Remark 4** *For smooth non-convex problems, we recover the optimal $O(\epsilon^4)$ classical rate as vanilla SGD.*

## 6 Optimal Rate and Communication

In this section, we provide the "optimal" setting of our algorithm that is obtained by optimizing the complexity bounds of our algorithm as a function of the parameters involved. The analysis on this section is based on

Table 1: **Gradient compression methods used in this work. Note that $\|\tilde{g}\|_0$ and $\|g\|_0$ are the number of elements in the compressed and uncompressed gradient, respectively; nature of operator $\mathcal{C}$ is random or deterministic. We implement that mechanisms for FedML.ai framework.**

| Compression | Ref. | Similar Methods | $\|\tilde{g}\|_0$ | Nature of $\mathcal{C}$ |
|---|---|---|---|---|
| QSGD | Alistarh et al. (2017) | Horváth et al. (2019); Wang et al. (2018); Wen et al. (2017) Wu et al. (2018); Yu et al. (2019b); Zhang et al. (2017) | $\|g\|_0$ | Rand, unbiased |
| Natural | Horváth et al. (2019) | Alistarh et al. (2017); Yu et al. (2019b); Zhang et al. (2017) | $\|g\|_0$ | Rand, unbiased |
| TernGrad | Wen et al. (2017) | Alistarh et al. (2017); Wang et al. (2018); Yu et al. (2019b) | $\|g\|_0$ | Rand, unbiased |
| Bernoulli | Khirirat et al. (2018) | — | — | Rand, unbiased |
| Top-$k$ | Aji & Heafield (2017) | Alistarh et al. (2018); Stich et al. (2018) | $k$ | Det, Biased |

the following upper bound of $\gamma$. We recall that

$$
\begin{aligned}
\gamma &= \frac{\alpha\lambda^2(1-p)}{2n^2 p} + \max\left\{\frac{L_f}{(1-p)}, \frac{\lambda}{n}\left(1 + \frac{4(1-p)}{p}\right)\right\} \\
&\leq \frac{\alpha\lambda^2(1-p)}{2n^2 p} + \max\left\{\frac{L_f}{(1-p)}, \frac{4\lambda}{np}\right\} := \gamma_u.
\end{aligned}
$$

Note that the number of iterations is linearly dependent on $\gamma$. Therefore, to minimize the total number of iterations, it suffices to minimize $\gamma$. Define $L := nL_f$.

**Theorem 3 (Optimal rate)** *The probability $p^*$ minimizing $\gamma$ is equal to $\max\{p_e, p_A\}$, where $p_e = \frac{7\lambda+L-\sqrt{\lambda^2+14\lambda L+L^2}}{6\lambda}$ and $p_A$ is the optimizer of the function $A(p) = \frac{\alpha\lambda^2}{2n^2 p} + \frac{L}{n(1-p)}$ in $(0,1)$.*

**Remark 5** *If we maximize the upper bound, $\gamma_u$ instead of $\gamma$ then $p_e$ simplifies to $\frac{4\lambda}{L+4\lambda}$.*

**Lemma 7** *The optimizer probability $p_A$ of the function $A(p) = \frac{\alpha\lambda^2}{2n^2 p} + \frac{L}{n(1-p)}$ in $(0,1)$ is equal to*

$$
p_A = \begin{cases}
\frac{1}{2} & \text{if } 2nL = \alpha\lambda^2 \\
\frac{-2\alpha\lambda^2 + 2\lambda\sqrt{2\alpha nL}}{2(2nL - \alpha\lambda^2)} & \text{if } 2nL > \alpha\lambda^2 \\
\frac{-2\alpha\lambda^2 - 2\lambda\sqrt{2\alpha nL}}{2(2nL - \alpha\lambda^2)} & \text{otherwise.}
\end{cases}
$$

Note that the number of communication rounds is linearly proportional to $C := p(1-p)\gamma$. Therefore, minimizing the total number of communication rounds suffices to minimize $C$ or $nC$.

**Theorem 4 (Optimal communication)** *The probability $p^*$ optimizing $C$ is equal to $\max\{p_e, p_A\}$, where $p_e = \frac{7\lambda+L-\sqrt{\lambda^2+14\lambda L+L^2}}{6\lambda}$ and $p_A = 1 - \frac{Ln}{\alpha\lambda^2}$.*

**Remark 6** *As in Remark 5, we note that, if we use the upper bound, $\gamma_u$ instead of $\gamma$ then $p_e$ simplifies to $\frac{4\lambda}{L+4\lambda}$.*

We note that $\lambda \to 0$ implies $p^* \to 0$. This means that the optimal strategy, in this case, is *no communication at all*. This result is intuitive since for $\lambda = 0$, we deal with pure local models which can be computed without any communication. As $\lambda \to \infty$ implies $p^* \to 1$ denoting that the optimal strategy is to communicate often to find the global model.

## 7 Empirical Study

We conducted diverse numerical experiments with L2GD algorithm that includes: ($i$) Analysis of algorithm meta-parameters (with and without compression) for logistic regression in strongly convex setting; see §7.1; ($ii$) analysis of compressed L2GD algorithm on image classification with DNNs; see §7.2.

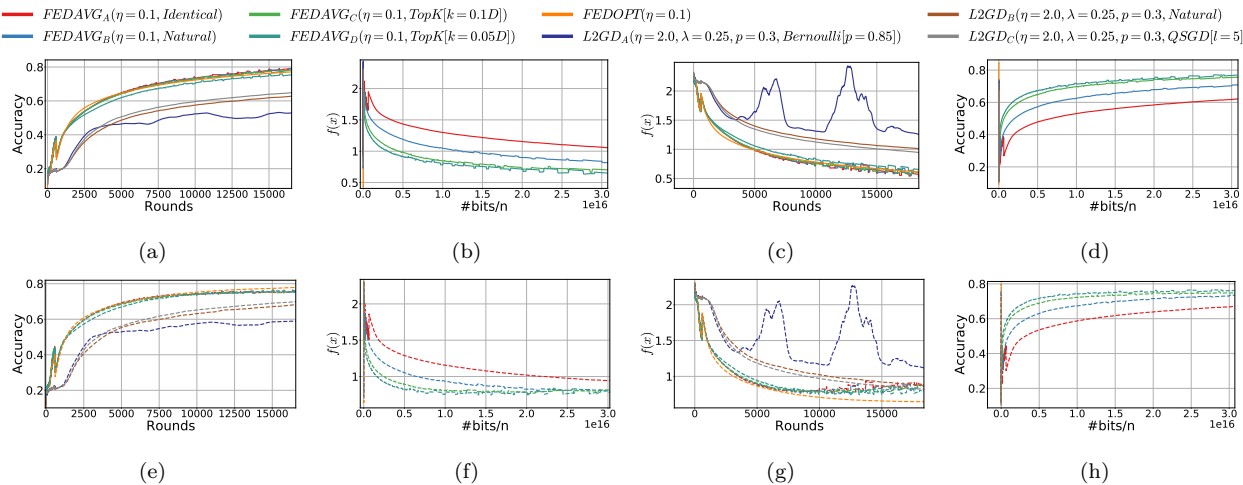

**Figure 4: Training `ResNet-18` on `CIFAR-10` with** $n = 10$ **workers. The top row represents the Top-1 accuracy vs. rounds in (a), loss functional value vs. communicated bits in (b), loss functional value vs. rounds in (c), and Top-1 accuracy vs. communicated bits in (d) on the train set. The bottom row presents the similar plots on the Test set in (e)−(h).**

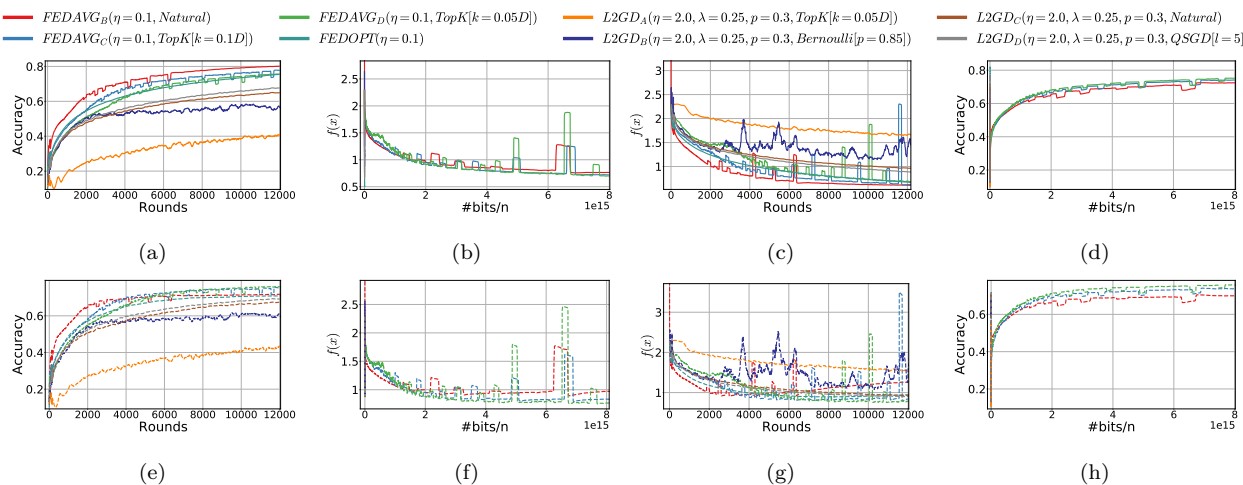

**Figure 5: Training `DenseNet-121` on `CIFAR-10` with** $n = 10$ **workers. The top row represents the Top-1 accuracy vs. rounds in (a), loss functional value vs. communicated bits in (b), loss functional value vs. rounds in (c), and Top-1 accuracy vs. communicated bits in (d) on the train set. The bottom row presents the similar quantities on the Test set in (e)−(h).**

**Computing environment.** We performed experiments on server-grade machines running Ubuntu 18.04 and Linux Kernel v5.4.0, equipped with 8-cores 3.3 GHz Intel Xeon and a single NVIDIA GeForce RTX 2080 Ti.Tesla-V100-SXM2 GPU with 32GB of GPU memory. The computation backend for Logistics Regression experiments was NumPy library with leveraging MPI4PY for inter-node communication. For DNNs we used recent version of FedML He et al. (2020) benchmark[4] and patched it with: (*i*) distributed and standalone version of Algorithm 1; (*ii*) serializing and plotting mechanism; (*iii*) modifications in standalone, distributed version of FedAvg McMahan et al. (2017) and FedOpt Reddi et al. (2020) to be consistent with equation 1; (*iv*) not to drop the last batch while processing the dataset.

---

[4]FedML.AI

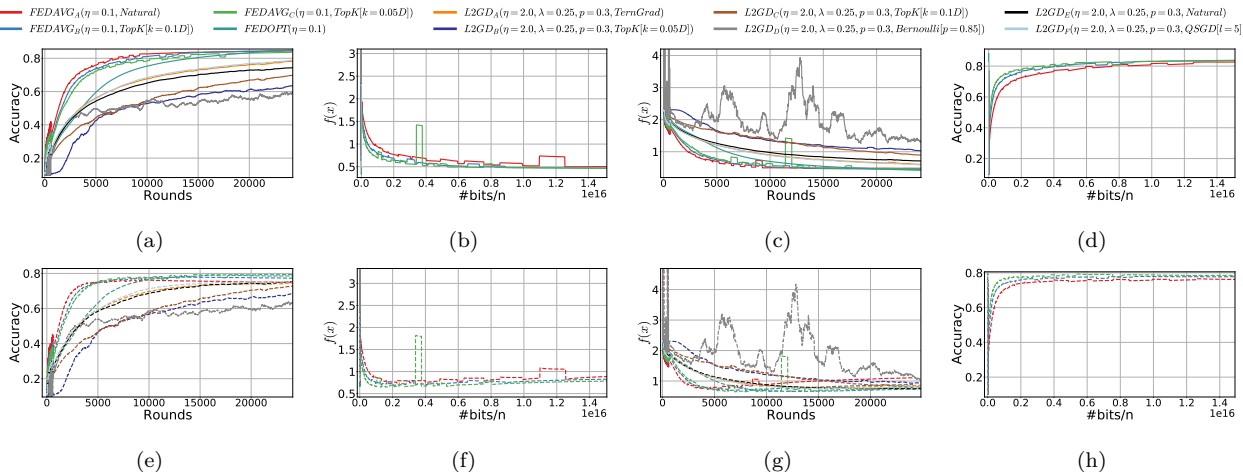

**Figure 6: Training MobileNet on CIFAR-10 with $n = 10$ workers. The top row represents the Top-1 accuracy vs. rounds in (a), loss functional value vs. communicated bits in (b), loss functional value vs. rounds in (c), and Top-1 accuracy vs. communicated bits in (d) on the train set. The bottom row presents the similar plots on the Test set in (e)−(h).**

## 7.1 Meta-parameter study

The purpose of these experiments is to study the meta-parameters involved in uncompressed and compressed L2GD algorithm. We used L2GD algorithm with and without compression for solving $\ell_2$ regularized logistic regression on LIBSVM `a1a` and `a2a` datasets Chang & Lin (2011). Both datasets contain shuffled examples in the train set, and we did not perform any extra shuffling. To simulate the FL settings, we divided both datasets into 5 parts. After splitting, each worker has 321 and 453 records for `a1a`, and `a2a`, respectively.

**Setup and results.** We define $f_i(x)$ to be local empirical risk minimization for logistic loss with additive regularization term for local data $D_i$ and of the form:

$$f_i(x) = \frac{1}{n_i} \sum_{j=1}^{n_i} \log(1 + \exp(-b^{(j)} x^\top a^{(j)})) + \frac{L_2}{2} \|x\|^2,$$

where $a^{(j)} \in \mathbb{R}^{124}, b^{(j)} \in \{+1, -1\}, n_i = |D_i|$. We set $L_2 = 1$, and varied meta-parameters $p$ and $\lambda$. For each parameter, we performed 100 iterations of Algorithm 1. Note that, as meta-parameter $\lambda$ decreases the models will fit more to its local data, while $p$ provides stochastic balance between local gradient steps with probability, $1 - p$ and aggregation with probability, $p$. For compressed L2GD, we used natural compressor at the workers and identity compressor at the master.

**Takeaway message.** The results in Figure 2 support the theoretical finding—there exists an optimal choice of $(p, \lambda)$, where the loss function, $f$ achieves the least value. Nevertheless, this choice is problem dependent. Additionally, we find small $p$ is not good due to lack of samples in a single node compared to samples available at other nodes. There is a trade-off for each node in learning from the other nodes' data and spending time to learn from its own data. For uncompressed L2GD, the optimal setting of our algorithm is attained for $p = 0.4$ and $\lambda$ in $[0, 25]$. The same observations hold for compressed L2GD; see Figure 3. For compressed L2GD with natural compressor, the optimal setting of our algorithm is attained for $p = 0.8$ and $\lambda$ in $[0, 5]$. Finally, we observe that *to get the smallest errors on the training and validation sets, it is better not to perform the averaging step too often.*

## 7.2 Training DNN models

We choose four practically important DNN models used for image classification, and other down-streaming tasks, such as feature extractions for image segmentation, object detection, image embedding, image captioning, to name a few.

- **ResNet** (He et al., 2016). The overwhelmingly popular `ResNet` architecture exploits residual connections to remedy vanishing gradients. The network supported the trend toward smaller filters and deeper architectures, more curated towards FL training. We use `ResNet-18` and `ResNet-56` architectures (He et al., 2016).

- **DenseNet** (Huang et al., 2017) contains a short connection between layers via connecting each layer to every other layer in a feed-forward fashion. Dense connection allows propagating information to the final classifier via concatenating all feature maps. Each layer in `DenseNet` is narrow and contains only 12 filters—another practical model for FL training.

- **MobileNet** (Howard et al., 2017). DNN architecture has a trade off between computational complexity and accuracy (Bianco et al., 2018, p.3, Fig.1). For mobile devices that appear in cross-device FL, the computation cost and energy consumption are both important. The energy consumption is mostly driven by memory movement (Chen et al., 2018b; Horowitz, 2014). In `MobileNet` architecture standard convolution blocks performs depth-wise convolution followed by $1 \times 1$ convolution. This is computationally less expensive in flops during inference time (see (Bianco et al., 2018, Fig.1, p.3)) and is $\sim 3.5\times$ more power efficient compare to `DenseNet` (García-Martín et al., 2019, p.85, Table 7). This makes `MobileNet` an attractive model for FL training.

**Dataset and setup.** We consider `CIFAR-10` dataset (Krizhevsky & Hinton, 2009) for image classification. It contains color images of resolution $28 \times 28$ from 10 classes. The training and the test set are of size, $5 \times 10^4$ and $10^4$, respectively. The training set is partitioned heterogeneously across 10 clients. The proportion of samples of each class stored at each local node is drawn by using the Dirichlet distribution ($\alpha = 0.5$). In our experiments, all clients are involved in each communication round. Additionally, we added a linear head in all CNN models for `CIFAR-10`, as they are originally designed for classification task with 1000 output classes.

**Loss function.** Denote $f_i(x) = w_i \cdot \frac{1}{|D_i|} \sum_{(a_i, b_i) \in D_i} l(a_i, b_i, x)$ to be a weighted local empirical risk associated with the local data, $\mathcal{D}_i$ stored in node, $i$. We note that $l(a_i, b_i, x)$ is a standard unweighted cross-entropy loss, $a_i \in \mathbb{R}^{28 \times 28 \times 3}$, $b_i \in \{0, 1\}^{10}$ with only one component equal to 1, the ground truth value, and the weight is set to $w_i = |D_i|/|D_1 \cup \cdots \cup D_n|$.

**Metrics.** To measure the performance, we examine the loss function value, $f(x)$, and the Top-1 accuracy on both train and the test set. We use the weighted average of the local models, where the weight, $w_i$ for each client is defined above. In our experiments, we wanted to assess the efficiency of both models — the local models and the global model (or the average model), $\frac{1}{n} \sum x_i$. To do this, we used two metrics:

- **Loss.** We compute the average loss over all the losses of the local models, that is, $f(x_1, \cdots, x_n) = \frac{1}{n} \sum_i f_i(x_i)$. This allows us to assess the efficiency of the local models.

- **Accuracy.** We compute the accuracy of the average model, that is, we compute the accuracy using the model. As the experiments demonstrate both local models and the average model perform well.

We use state-of-the-art FedML benchmarking for our experiments, and it does not support personalization. Due to this limitation, although we wanted to present the average accuracy over all the accuracies of the local models, we were unable to do so — changing the FedML benchmarking for personalization is an involved task. Nevertheless, we compared with compressed communication approaches in the FedML framework that supports a single global model to see how despite personalization, our compressed L2GD performs. Additionally, we measure the number of rounds, and bits/$n$—communicated bits normalized by the number of local clients, $n$. The intuition behind using the last metric is to measure the communicated data-volume; it is widely *hypothesized* that the reduced data-volume translates to a faster training in a constant speed network in distributed setup (Gajjala et al., 2020; Xu et al., 2021a).

**Compressors used.** The theoretical results of compressed L2GD are attributed to unbiased compressors. We used 4 different unbiased compressors at the clients: Bernoulli (Khirirat et al., 2018), natural compressor (Horváth et al., 2019), random dithering a.k.a. QSGD (Alistarh et al., 2017), and Terngrad (Wen et al., 2017); see Table 1 for details. Additionally, we note that biased compressors (mostly sparsifiers) are popular in DNN

| Model | Training parameters | L2GD bits/$n$ | Baseline bits/$n$ |
|---|---|---|---|
| DenseNet-121 | $79 \times 10^5$ | $8 \times 10^{11}$ | $4 \cdot 10^{15}$ |
| MobileNet | $32 \times 10^5$ | $1.7 \times 10^{11}$ | $1 \times 10^{15}$ |
| ResNet-18 | $11 \times 10^6$ | $1.1 \times 10^{12}$ | $1.5 \times 10^{16}$ |

**Table 2: Summary of the benchmarks. The measured quantity is** bits/$n$ **to achieve** 0.7 **Top-1 test accuracy, with** $n = 10$ **clients. For DenseNet-121, MobileNet, Resnet-18 the baseline is FedAvg with natural compressor with** 1 **local epoch; L2GD also uses natural compressor.**

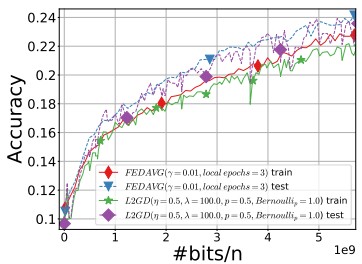

**Figure 7: FedAvg as a particular case of L2GD: Test and train accuracy for ResNet-56 on CIFAR-10.**

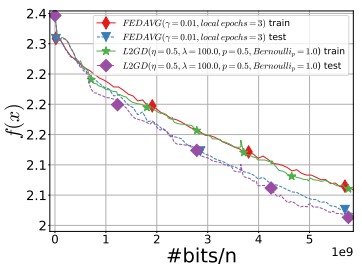

**Figure 8: FedAvg as a particular case of L2GD: Test and train loss for ResNet-56 on CIFAR-10.**

training. Therefore, out of scientific curiosity, we used a popular sparsifier: Top-$k$ (Aji & Heafield, 2017; Sahu et al., 2021) as a proof of concept. We note that extending the compressed L2GD theory for biased compressors (with or without error-feedback (Xu et al., 2021a)) is nontrivial and mathematically involved, and left for future work.

**Algorithms used for comparison.** We used state-of-the-art FL training algorithms, FedAvg (McMahan et al., 2017) and FedOpt (Reddi et al., 2020) as no compression baseline to compare against our L2GD. However, the performance of FedAvg is not stable but improves with the compression mechanism. The original FedAvg algorithm does not contain any compression mechanism, but for comparison, we incorporated compressors into FedAvg via the following schema which is similar to the classic error feedback (Xu et al., 2021a): ($i$) After local steps, client estimates change of current iterate from the previous round and formulates direction, $g_{c,\text{computed}}{}^i$; ($ii$) client sends compressed difference between previous gradient estimator from previous round and currently computed gradient estimator, $\mathcal{C}(g_{c,\text{computed}}{}^i - g_c{}^{i-1})$ to the master; ($iii$) both master and client updating $g_c{}^i$ via the following schema: $g_c{}^i = g_c{}^{i-1} + \mathcal{C}(g_{c,\text{computed}}{}^i - g_c{}^{i-1})$. We provide the details about step size and batch size in Appendix.

### 7.2.1 Results

We show the results for training `ResNet-18`, `DenseNet-121`, and `MobileNet` with compressed L2GD and other state-of-the-art FL algorithms in Figure 4–6. For these experiments, the communication rounds are set to $12 \times 10^3$, $25 \times 10^3$, and $20 \times 10^3$, respectively. For the FedAvg algorithm, each client performs one epoch over the local data. We empirically tried $1, 2, 3$, and $4$ epochs over the local data as local steps, but one epoch is empirically the best choice.

For training `ResNet-18`, from Figure 4 we observe that FedAvg with compression has albeit better convergence than no compression FedAvg [5]. At the same time, compressed FedAvg affects the convergence as a function of communicated rounds only negligibly (see Figure 4 (d),(b)). Therefore, for training other DNN models we use FedAvg with compression and FedOpt without any compressors to enjoy the best of both baselines.

**Take away message.** Compressed L2GD with natural compressor sends the least data and drives the loss down the most in these experiments. At the same time, L2GD with natural compressor (by design it has smaller variance) reaches the best accuracy for both train and test sets. Compressed L2GD outperforms FedAvg by a huge margin—For all DNN experiments, to reach the desired Top-1 test accuracy, compressed L2GD reduces the communicated data-volume, #bits/n, from $10^{15}$ to $10^{11}$, rendering approximately a $10^4$ times improvement compared to FedAvg; see Table 2.

---

[5]We have observed that batch normalization (Ioffe & Szegedy, 2015) in ResNet is sensitive for aggregation; see our discussion in §A.2.

Interestingly, in training `MobileNet`, the performance of biased Top-$k$ compressor degrades only about 10% compared to natural compressor, while approximately degrades 35% in training `DenseNet`. Additionally, see discussion in §A.2, Figures 9–11. This phenomena may lead the researchers to design unbiased compressors with smaller variance to empirically harvest the best behavior of compressed L2GD in personalized FL training.

Nevertheless, we also observe that compressed L2GD converges slower compared to other FL algorithms without compression in all cases. What follows, it can be argued, is that when we compare the communicated data volume for all DNN models, the convergence of compressed L2GD is much better. Additionally, the gain in terms of lowering the loss function value is significant—*by sending the same amount of data, L2GD lowers the loss the most compared to the other no-compression FL baseline algorithms.* These experiments also demonstrate that when communication is a bottleneck, FedAvg is not comparable with L2GD. The only comparable baseline for L2GD is FedOpt; see Table II, also, see discussion in §A.2, Figures 9–11. A similar observation holds for the Top-1 test and train accuracy. Taken together, these indicate that for training larger DNN models in a personalized FL settings, with resource constrained and geographically remote devices, compressed L2GD could be the preferred algorithm because its probabilistic communication protocol sends less data but obtains better test accuracy than no compression FedAvg and FedOpt.

Additionally, we observe that when $\frac{\eta\lambda}{np} \in [0.5, 0.95]$, compressed L2GD incurs a significant variance in objective function during training. Empirically, the best behavior was observed for $\frac{\eta\lambda}{np} \approx 1$ or $\frac{\eta\lambda}{np} \in (0, 0.17]$.

**FedAvg as a particular case of L2GD.** We note that if $\eta\lambda/np = 1$, then the aggregation step of Algorithm 1 reduces to $x_i^{k+1} = \bar{x}^k$, for all devices. Thus, in this regime L2GD works similarly as FedAvg with random number of local steps. E.g.,if $p = 0.5$, then Algorithm 1 reduces to randomized version of FedAvg with an average of 3 local steps. Figures 7 and 8, confirm this observation numerically, where we see that both algorithms exhibit similar performance. For that experiment, we trained `ResNet-56` on `CIFAR10` with $n = 100$ workers, and for 600 rounds. For `L2GD`, we set $\frac{\eta\lambda}{pn} = 1$.

## 8 Conclusion and Future Direction

In this paper, we equipped the loopless gradient descent (L2GD) algorithm with a compression mechanism to reduce the communication bottleneck between local devices and the server in an FL context. We showed that the new algorithm enjoys similar convergence properties as the uncompressed L2GD with a natural increase in the stochastic gradient variance due to compression. This phenomenon is similar to classical convergence bounds for compressed SGD algorithms. We also show that in a personalized FL setting, there is a trade-off that must be considered by devices between learning from other devices' data and spending time learning from their own data. However, a particular parameterization of our algorithm recovers the well-known FedAvg Algorithm. We assessed the performance of the new algorithm compared to the state-of-the-art and validated our theoretical insights through a large set of experiments.

Several questions remain open and merit further investigation in the future. For example, we plan on including compression when devices calculate their local updates, especially in an FL setting, as the devices might not be powerful, and the computing energy is limited, and examine how the algorithm behaves. Additionally, we observed the efficacy of compressed L2GD with a biased compressor, such as Top-$k$. Nevertheless, extending the compressed L2GD theory for biased compressors (with or without error-feedback (Xu et al., 2021a)) is nontrivial and challenging. In the future, we plan to prove a more general theory for compressed L2GD that include both biased and unbiased compressor operating in a bidirectional fashion. A more detailed meta-parameter study covering different network bandwidths, diverse ML tasks with different DNN architectures, and deploying the models on real-life, geographically remote servers will be our future empirical quest.

## Acknowledgments

Aritra Dutta acknowledges being an affiliated researcher at the Pioneer Centre for AI, Denmark. The authors acknowledge many fruitful discussions with Md. Patel on this project while he was a remote undergraduate intern at KAUST.

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

# A  Appendix

## Contents

### A.1  Convergence Analysis—Proofs of the Lemmas and the Theorems

In this section, we provide the proofs of convex and non-convex convergence results of the compressed L2GD algorithm.

**Overview of results.** In §A.1.1, we provide the technical lemmas necessary for the analyses. §A.1.2 contains the auxiliary results pertaining to both convex and nonconvex convergence. In §A.1.3 we provide the non-convex convergence results, and §A.1.4 contains the proofs for optimal rate and communication.

### A.1.1 Technical results used for convergence

The following two Lemmas are instrumental in proving other compression related results.

**Lemma 1** *Let $x \in R^{nd}$, then*

$$\mathbb{E}_{\mathcal{C}} \left[ \|\mathcal{C}(x)\|^2 \right] \leq (1+\omega)\|x\|^2,$$

*where $\omega = \max_{i=1,\ldots,n}\{\omega_i\}$.*

**Proof 2** *By using Assumption 1, we have*

$$\mathbb{E}_{\mathcal{C}} \left[ \|\mathcal{C}(x)\|^2 \right] \quad = \quad \mathbb{E}_{\mathcal{C}} \left[ \sum_{i=1}^n \|\mathcal{C}_i(x_i)\|^2 \right] = \sum_{i=1}^n \mathbb{E}_{\mathcal{C}_i} \|\mathcal{C}_i(x_i)\|^2 \leq \sum_{i=1}^n (1+\omega_i)\|x_i\|^2 \leq (1+\omega)\|x\|^2.$$

*Hence the result.*

**Lemma 2** *Let Assumption 1 hold, then for all $k \geq 0$, $\mathbb{E}_{\mathcal{C},\mathcal{C}_M} \left[ \mathcal{C}_M(\bar{y}^k) \right] = \bar{x}^k$.*

**Proof 3** *We have*

$$\mathbb{E}_{\mathcal{C},\mathcal{C}_M} \left[ \mathcal{C}_M(\bar{y}^k) \right] = \mathbb{E}_{\mathcal{C}} \left[ \mathbb{E}_{\mathcal{C}_M} \left[ \mathcal{C}_M(\bar{y}^k) \right] \right] = \mathbb{E}_{\mathcal{C}} \left[ \frac{1}{n} \sum_{j=1}^n \mathcal{C}_j(x_j^k) \right] = \frac{1}{n} \sum_{j=1}^n \mathbb{E}_{\mathcal{C}_j} \left[ \mathcal{C}_j(x_j^k) \right] = \bar{x}^k.$$

*Hence the result.*

In the following Lemma, we show that based on the randomness of the compression operators, in expectation, we recover the exact average of the local models and the exact gradient for all iterations.

**Lemma 3** *Let Assumptions 1 hold. Then for all $k \geq 0$, knowing $x^k$, $G(x^k)$ is an unbiased estimator of the gradient of function $F$ at $x^k$.*

**Proof 4** *We have*

$$\mathbb{E}_{\mathcal{C},\mathcal{C}_M} \left[ G_i(x^k) \right] \quad = \quad \begin{cases} \frac{\nabla f_i(x_i^k)}{n(1-p)} & if \; \xi_k = 0 \\ \frac{\lambda}{np} \left( x_i^k - \mathbb{E}_{C,\mathcal{C}_M} \left[ \mathcal{C}_M(\bar{y}^k) \right] \right) & if \; \xi_k = 1 \; \& \; \xi_{k-1} = 0, \\ \frac{\lambda}{np} \left( x_i^k - \bar{x}^k \right) & if \; \xi_k = 1 \; \& \; \xi_{k-1} = 1, \end{cases}$$

$$\stackrel{\text{By Lemma 2}}{=} \quad \begin{cases} \frac{\nabla f_i(x_i^k)}{n(1-p)} & if \; \xi_k = 0, \\ \frac{\lambda}{np} \left( x_i^k - \bar{x}^k \right) & if \; \xi_k = 1. \end{cases}$$

*Therefore,*

$$\begin{aligned} \mathbb{E}[G_i(x^k)|x^k] \quad &= \quad \mathbb{E}_{\xi_k} \left[ \mathbb{E}_{C,\mathcal{C}_M} \left[ G_i(x^k) \right] \right] \\ &= \quad (1-p)\frac{\nabla f_i(x_i^k)}{n(1-p)} + p\frac{\lambda}{np} \left( x_i^k - \bar{x}^k \right) \\ &= \quad \nabla_{x_i} f(x^k) + \nabla_{x_i} h \left( x^k \right) = \nabla_{x_i} F(x^k). \end{aligned}$$

*Hence the result.*

### A.1.2  Main convergence results

Based on the results given in the previous section, we are now set to quote our key convergence results. Our next lemma gives an upper bound on the iterate at each iteration. This bound is composed of two terms—the optimality gap, $F(x^k) - F(x^*)$, and the norm of the optimal point, $\|x^*\|$.

**Lemma 8** *For a $\mu$-strongly convex function $F$, we have $\|x - x^*\| \leq \frac{2}{\mu}(F(x) - F(x^*))$.*

**Proof 5** *For a $\mu$-strongly convex function, $F$, for all $x, y$ we have*

$$F(x) \geq F(y) + \nabla F(y)^\top (x - y) + \frac{\mu}{2}\|x - y\|^2.$$

*At the optimal point $y = x^*$, we have $\nabla F(x^*) = 0$, and we obtain the desired result.*

**Lemma 4** *Let Assumption 2 hold, then*

$$\left\|x^k\right\|^2 \leq \tfrac{4}{\mu}\left(F(x^k) - F(x^*)\right) + 2\left\|x^*\right\|^2.$$

**Proof 6** *We have*

$$\left\|x^k\right\|^2 \overset{\|a+b\|^2 \leq 2\|a\|^2 + 2\|b\|^2}{\leq} 2\left\|x^k - x^*\right\|^2 + 2\left\|x^*\right\|^2 \overset{\text{By Lemma8}}{\leq} \frac{4}{\mu}\left(F(x^k) - F(x^*)\right) + 2\left\|x^*\right\|^2.$$

*Hence the result.*

Recall that, inspired by the expected smoothness property Gower et al. (2019), we use a similar idea in our convergence proofs. The next lemma is a technical Lemma that helps us to prove the expected smoothness property Gower et al. (2019). The bound in Lemma 5 is composed of the optimality gap, $F(x^k) - F(x^*)$, the difference between the gradients of $h$ at $x^k$ and $x^*$, that is, $\left\|\nabla h(x^k) - \nabla h(x^*)\right\|$, and an extra constant, $\beta$, which depends on the used compressors.

**Lemma 5** *Let Assumptions 1 and 2 hold, then*

$$\mathcal{A} := \mathbb{E}_{\mathcal{C}_M,\mathcal{C}}\left\|x^k - Q\mathcal{C}_M(\bar{y}^k) - x^* + Q\mathcal{C}_M(\bar{y}^*)\right\|^2 \leq \tfrac{4n^2}{\lambda^2}\left\|\nabla h(x^k) - \nabla h(x^*)\right\|^2 + \alpha\left(F(x^k) - F(x^*)\right) + \beta,$$

*where $\bar{y}^* := \frac{1}{n}\sum_{j=1}^{n}\mathcal{C}_j(x_j^*)$, $\alpha := \frac{4(4\omega + 4\omega_M(1+\omega))}{\mu}$, and*

$$\beta := 2\left(4\omega + 4\omega_M(1+\omega)\right)\|x^*\|^2 + 4\mathbb{E}_{\mathcal{C}_M,\mathcal{C}}\left\|Q\mathcal{C}_M(\bar{y}^*) - Q\bar{x}^*\right\|^2.$$

**Proof 7** *We have*

$$
\begin{aligned}
\mathcal{A} \quad &= \quad \mathbb{E}_{\mathcal{C}_M,\mathcal{C}} \left\| x^k - Q\bar{x}^k + Q\bar{x}^k - Q\mathcal{C}_M(\bar{y}^k) - x^* + Q\bar{x}^* - Q\bar{x}^* + Q\mathcal{C}_M(\bar{y}^*) \right\|^2 \\
&= \quad \mathbb{E}_{\mathcal{C}_M,\mathcal{C}} \left\| (x^k - Q\bar{x}^k - x^* + Q\bar{x}^*) + (Q\bar{x}^k - Q\bar{y}^k) + (Q\bar{y}^k - Q\mathcal{C}_M(\bar{y}^k)) + (Q\mathcal{C}_M(\bar{y}^*) - Q\bar{x}^*) \right\|^2 \\
&\leq \quad 4 \left\| x^k - Q\bar{x}^k - x^* + Q\bar{x}^* \right\|^2 + 4\mathbb{E}_{\mathcal{C}} \left\| Q\bar{x}^k - Q\bar{y}^k \right\|^2 + 4\mathbb{E}_{\mathcal{C}_M,\mathcal{C}} \left\| Q\bar{y}^k - Q\mathcal{C}_M(\bar{y}^k) \right\|^2 \\
&\quad + \quad 4\mathbb{E}_{\mathcal{C}_M,\mathcal{C}} \left\| Q\mathcal{C}_M(\bar{y}^*) - Q\bar{x}^* \right\|^2 \\
&= \quad 4 \left\| x^k - Q\bar{x}^k - x^* + Q\bar{x}^* \right\|^2 + 4n\mathbb{E}_{\mathcal{C}} \left\| \bar{x}^k - \bar{y}^k \right\|^2 + 4n\mathbb{E}_{\mathcal{C}_M,\mathcal{C}} \left\| \bar{y}^k - \mathcal{C}_M(\bar{y}^k) \right\|^2 \\
&\quad + \quad 4\mathbb{E}_{\mathcal{C}_M,\mathcal{C}} \left\| Q\mathcal{C}_M(\bar{y}^*) - Q\bar{x}^* \right\|^2 \\
&\leq \quad 4 \left\| x^k - Q\bar{x}^k - x^* + Q\bar{x}^* \right\|^2 + 4\sum_{i=1}^{n} \mathbb{E}_{\mathcal{C}} \left\| x_i^k - \mathcal{C}_i(x_i^k) \right\|^2 + 4n\omega_M \mathbb{E}_{\mathcal{C}_M} \left\| \bar{y}^k \right\|^2 \\
&\quad + \quad 4\mathbb{E}_{\mathcal{C}_M,\mathcal{C}} \left\| Q\mathcal{C}_M(\bar{y}^*) - Q\bar{x}^* \right\|^2 \\
&\leq \quad 4 \left\| x^k - Q\bar{x}^k - x^* + Q\bar{x}^* \right\|^2 + 4\sum_{i=1}^{n} \omega_i \left\| x_i^k \right\|^2 + 4\omega_M \sum_{i=1}^{n} (1 + \omega_i) \left\| x_i^k \right\|^2 + 4\mathbb{E}_{\mathcal{C}_M,\mathcal{C}} \left\| Q\mathcal{C}_M(\bar{y}^*) - Q\bar{x}^* \right\|^2 \\
&\leq \quad 4\frac{n^2}{\lambda^2} \left\| \nabla h(x^k) - \nabla h(x^*) \right\|^2 + (4\omega + 4\omega_M(1 + \omega)) \left\| x^k \right\|^2 + 4\mathbb{E}_{\mathcal{C}_M,\mathcal{C}} \left\| Q\mathcal{C}_M(\bar{y}^*) - Q\bar{x}^* \right\|^2 \\
\overset{\text{By Lemma 4}}{\leq} \quad &4\frac{n^2}{\lambda^2} \left\| \nabla h(x^k) - \nabla h(x^*) \right\|^2 + (4\omega + 4\omega_M(1 + \omega)) \left( \frac{4}{\mu} \left( F(x^k) - F(x^*) \right) + 2 \left\| x^* \right\|^2 \right) \\
&\quad + 4\mathbb{E}_{\mathcal{C}_M,\mathcal{C}} \left\| Q\mathcal{C}_M(\bar{y}^*) - Q\bar{x}^* \right\|^2 \\
&\leq \quad 4\frac{n^2}{\lambda^2} \left\| \nabla h(x^k) - \nabla h(x^*) \right\|^2 + \alpha \left( F(x^k) - F(x^*) \right) + \beta.
\end{aligned}
$$

*Hence the result.*

Now we are all set to prove the expected smoothness property in our setup.

**Lemma 6 (Expected Smoothness)** *Let Assumptions 1 and 2 hold, then*

$$
\mathbb{E}\left[ \|G(x^k)\|^2 | x^k \right] \leq 4\gamma \left( F(x^k) - F(x^*) \right) + \delta, \tag{3}
$$

*where*

$$
\gamma := \frac{\alpha\lambda^2(1-p)}{2n^2 p} + \max\left\{ \frac{L_f}{(1-p)}, \frac{\lambda}{n}\left(1 + \frac{4(1-p)}{p}\right) \right\}
$$

*and*

$$
\delta := \frac{2\beta\lambda^2(1-p)}{n^2 p} + 2\mathbb{E}\|G(x^*)\|^2.
$$

**Proof 8** *We have*

$$
\|G(x^k) - G(x^*)\|^2 = \begin{cases} \frac{\|\nabla f(x^k) - \nabla f(x^*)\|^2}{(1-p)^2} & \text{if } \xi_k = 0 \\ \frac{\lambda^2}{n^2 p^2} \left\| x^k - Q\mathcal{C}_M(\bar{y}^k) - x^* + Q\mathcal{C}_M(\bar{y}^*) \right\|^2 & \text{if } \xi_k = 1 \ \& \ \xi_{k-1} = 0, \\ \frac{1}{p^2} \|\nabla h(x^k) - \nabla h(x^*)\|^2 & \text{if } \xi_k = 1 \ \& \ \xi_{k-1} = 1. \end{cases}
$$

*Finally,*

$$
\begin{aligned}
\mathbb{E}_{\xi_k,\xi_{k-1}} \|G(x^k) - G(x^*)\|^2 &= (1-p)\frac{\|\nabla f(x^k) - \nabla f(x^*)\|^2}{(1-p)^2} + p^2 \frac{1}{p^2} \|\nabla h(x^k) - \nabla h(x^*)\|^2 \\
&\quad + p(1-p)\frac{\lambda^2}{n^2 p^2} \left\| x^k - Q\mathcal{C}_M(\bar{y}^k) - x^* + Q\mathcal{C}_M(\bar{y}^*) \right\|^2 \\
&= \frac{\|\nabla f(x^k) - \nabla f(x^*)\|^2}{(1-p)} + \|\nabla h(x^k) - \nabla h(x^*)\|^2 \\
&\quad + \frac{\lambda^2(1-p)}{n^2 p} \left\| x^k - Q\mathcal{C}_M(\bar{y}^k) - x^* + Q\mathcal{C}_M(\bar{y}^*) \right\|^2.
\end{aligned}
$$

*Therefore, by using Lemma 5 we get*

$$
\begin{aligned}
\mathbb{E}\|G(x^k) - G(x^*)|x^k\|^2 &= \frac{\|\nabla f\left(x^k\right) - \nabla f\left(x^*\right)\|^2}{(1-p)} + \|\nabla h(x^k) - \nabla h(x^*)\|^2 + \frac{\lambda^2(1-p)}{n^2 p}\mathcal{A} \\
&\leq \frac{\|\nabla f\left(x^k\right) - \nabla f\left(x^*\right)\|^2}{(1-p)} + \|\nabla h(x^k) - \nabla h(x^*)\|^2 \\
&\quad + \frac{\lambda^2(1-p)}{n^2 p}\left(4\frac{n^2}{\lambda^2}\left\|\nabla h(x^k) - \nabla h(x^*)\right\|^2 + \alpha\left(F(x^k) - F(x^*)\right) + \beta\right) \\
&= \frac{\|\nabla f\left(x^k\right) - \nabla f\left(x^*\right)\|^2}{(1-p)} + \left(1 + \frac{4(1-p)}{p}\right)\|\nabla h(x^k) - \nabla h(x^*)\|^2 \\
&\quad + \frac{\alpha\lambda^2(1-p)}{n^2 p}\left(F(x^k) - F(x^*)\right) + \frac{\beta\lambda^2(1-p)}{n^2 p} \\
&\leq \frac{2L_f}{(1-p)}\left(f(x^k) - f(x^*)\right) + \frac{2\lambda}{n}\left(1 + \frac{4(1-p)}{p}\right)\left(h(x^k) - h(x^*)\right) \\
&\quad + \frac{\alpha\lambda^2(1-p)}{n^2 p}\left(F(x^k) - F(x^*)\right) + \frac{\beta\lambda^2(1-p)}{n^2 p} \\
&\leq 2\gamma\left(F(x^k) - F(x^*)\right) + \frac{\beta\lambda^2(1-p)}{n^2 p}.
\end{aligned}
$$

*Finally, we obtain*

$$
\begin{aligned}
\mathbb{E}\|G(x^k)|x^k\|^2 &\leq 2\mathbb{E}\|G(x^k) - G(x^*)|x^k\|^2 + 2\mathbb{E}\|G(x^*)\|^2 \\
&\leq 4\gamma\left(F(x^k) - F(x^*)\right) + \frac{2\beta\lambda^2(1-p)}{n^2 p} + 2\mathbb{E}\|G(x^*)\|^2 \\
&\leq 4\gamma\left(F(x^k) - F(x^*)\right) + \delta.
\end{aligned}
$$

*Hence the result.*

Based on the above results, the convergence of Algorithm 1 for strongly convex functions follows directly from Lemmas 3, 6 and Theorem 3.1 from Gower et al. (2019).

### A.1.3  Nonconvex convergence

**Theorem 5 (Non convex case)** *Let Assumptions 1 and 3 hold. Assume also that $F$ is $L_f$-smooth, bounded from below by $F(x^*)$. Then to reach a precision, $\epsilon > 0$, set the stepsize, $\eta = \min\{\frac{1}{L_f M}, \frac{\epsilon^2}{2L_f \sigma^2}\}$, such that for $K \geq \frac{4L_f M(F(x^0) - F(x^*))}{\epsilon^2}$, we have $\min_{k=0,1,\ldots,K}\mathbb{E}\|\nabla F(x^k)\|_2 \leq \epsilon$.*

**Proof 9** *From $L_f$-smoothness of $F$ we have*

$$
F(x^{k+1}) \leq F(x^k) - \eta_k \nabla F(x^k)^\top G(x^k) + \frac{L_f}{2}\eta_k^2 \|G(x_k)\|^2.
$$

*By taking the expectation in the above inequality, conditional on $x^k$, we get*

$$
\mathbb{E}\left[F(x^{k+1}) \mid x^k\right] \overset{\text{By Lemma 3}}{\leq} F(x^k) - \eta_k \|\nabla F(x^k)\|_2^2 + \frac{L_f \eta_k^2}{2}\mathbb{E}\left(\|G(x_k)\|^2 | x_k\right),
$$

*which by using Assumption 3 reduces to*

$$
\begin{aligned}
\mathbb{E}\left[F(x^{k+1}) \mid x^k\right] &\leq F(x^k) - \eta_k\|\nabla F(x^k)\|_2^2 + \frac{L_f \eta_k^2}{2}\left(M\|\nabla F(x^k)\|^2 + \sigma^2\right) \\
&\leq F(x^k) - \eta_k\left(1 - \frac{L_f M \eta_k}{2}\right)\|\nabla F(x^k)\|_2^2 + \frac{L_f \eta_k^2 \sigma^2}{2}.
\end{aligned}
$$

*After rearranging, we have*

$$\eta_k \left(1 - \frac{L_f M \eta_k}{2}\right) \|\nabla F(x^k)\|_2^2 \leq F(x^k) - \mathbb{E}\big[F(x^{k+1}) \mid x^k\big] + \frac{L_f \eta_k^2 \sigma^2}{2}.$$

*Setting $\eta_k = \eta > 0$ in the above, taking expectation, using the tower property of expectation, and finally summing over the iterates $k = 0, 1, \cdots K - 1$ we have*

$$\eta \left(1 - \frac{L_f M \eta}{2}\right) \sum_{k=0}^{K-1} \mathbb{E}\left[\|\nabla F(x^k)\|_2^2\right] \leq \big(F(x^0) - F(x^*)\big) + \frac{K L_f \eta \sigma^2}{2}.$$

*If $\eta \leq \frac{1}{L_f M}$ then*

$$\sum_{k=0}^{K-1} \mathbb{E}\left[\|\nabla F(x^k)\|_2^2\right] \leq \frac{2}{\eta}\big(F(x^0) - F(x^*)\big) + L_f K \eta \sigma^2.$$

*Dividing throughout by $K$, we get*

$$\frac{1}{K} \sum_{k=0}^{K-1} \mathbb{E}\left[\|\nabla F(x^k)\|_2^2\right] \leq \frac{2}{\eta K}\big(F(x^0) - F(x^*)\big) + L_f \eta \sigma^2.$$

*Finally, setting $\eta = \frac{1}{L_f M}$ we have*

$$\min_{k=0,1,\cdots K-1} \mathbb{E}\|\nabla F(x^k)\|^2 \leq \frac{2 L_f M}{K}\big(F(x^0) - F(x^*)\big) + \frac{\sigma^2}{M}. \tag{4}$$

*For a given precision, $\epsilon > 0$, to make $\min_{k=0,1,\cdots K-1} \mathbb{E}\|\nabla F(x^k)\|^2 \leq \epsilon^2$, we require $\frac{2 L_f M\big(F(x^0) - F(x^*)\big)}{K} \leq \frac{\epsilon^2}{2}$ and $L_f \eta \sigma^2 \leq \frac{\epsilon^2}{2}$, resulting in*

$$K \geq \frac{4 L_f M (F(x^0) - F(x^*))}{\epsilon^2} \text{ and } \eta \leq \frac{\epsilon^2}{2 L_f \sigma^2}.$$

*Hence the result.*

### A.1.4 Optimal rate and communication

The following proofs are related to optimal rate and communication as given in §6.

**Theorem 1 (Optimal rate)** *The probability $p^*$ minimizing $\gamma$ is equal to $\max\{p_e, p_A\}$, where $p_e = \frac{7\lambda + L - \sqrt{\lambda^2 + 14\lambda L + L^2}}{6\lambda}$ and $p_A$ is the optimizer of the function $A(p) = \frac{\alpha\lambda^2}{2n^2 p} + \frac{L}{n(1-p)}$ in $(0, 1)$.*

**Proof 10** *We can rewrite $\gamma$ as follows*

$$\gamma = -\frac{\alpha\lambda^2}{2n^2} + \max\{A(p), B(p)\},$$

*where $A(p) = \frac{\alpha\lambda^2}{2n^2 p} + \frac{L}{n(1-p)}$ and $B(p) = \frac{\alpha\lambda^2}{2n^2 p} + \frac{4\lambda}{np} - \frac{3\lambda}{n}$. The function $B$ is monotonically decreasing as a function of $p$. The function $A$ goes to $\infty$ as $p$ goes to zero or one, and it has one stationary point between zero and one hence it is convex in the interval $(0, 1)$. Thus it admits an optimizer $p_A$ in $(0, 1)$. Note that $p_e = \frac{7\lambda + L - \sqrt{\lambda^2 + 14\lambda L + L^2}}{6\lambda}$ is the point for which $A(p)$ is equal to $B(p)$. Note also that near to zero $B(p) \geq A(p)$. Therefore if $p_e \leq p_A$ then the optimizer of $\gamma$ is $p_A$ otherwise it is equal to $p_e$. Thus the probability $p^*$ optimizing $\gamma$ is equal to $\max\{p_e, p_A\}$.*

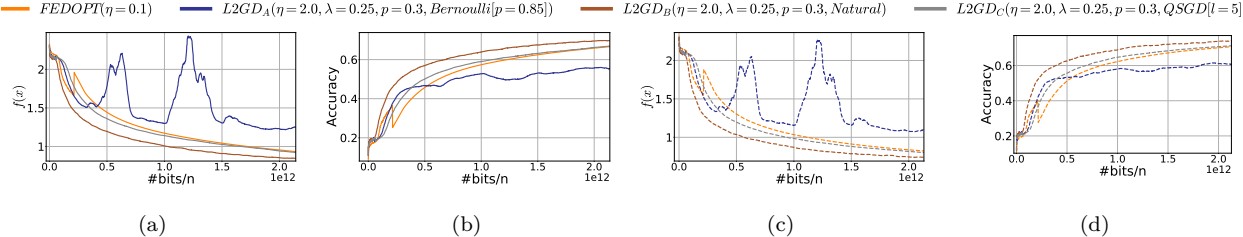

**Figure 9:** Training `ResNet-18` on `CIFAR-10`, with $n = 10$ **workers. Loss and Top-1 accuracy on train (a) - (b) and test data (c) - (d).**

**Lemma 7** *The optimizer probability $p_A$ of the function $A(p) = \frac{\alpha \lambda^2}{2n^2 p} + \frac{L}{n(1-p)}$ in $(0,1)$ is equal to*

$$
p_A = \begin{cases}
\frac{1}{2} & \text{if } 2nL = \alpha \lambda^2 \\
\frac{-2\alpha \lambda^2 + 2\lambda \sqrt{2\alpha n L}}{2(2nL - \alpha \lambda^2)} & \text{if } 2nL > \alpha \lambda^2 \\
\frac{-2\alpha \lambda^2 - 2\lambda \sqrt{2\alpha n L}}{2(2nL - \alpha \lambda^2)} & \text{otherwise}
\end{cases}
$$

**Proof 11** *If $2nL \neq \alpha \lambda^2$, then the function $A$ has the following two stationary points $\frac{-2\alpha \lambda^2 + 2\lambda \sqrt{2\alpha n L}}{2(2nL - \alpha \lambda^2)}$ and $\frac{-2\alpha \lambda^2 - 2\lambda \sqrt{2\alpha n L}}{2(2nL - \alpha \lambda^2)}$. If $2nL = \alpha \lambda^2$, then the function $A$ has one stationary point equal to $\frac{1}{2}$.*

**Theorem 2 (Optimal communication)** *The probability $p^*$ optimizing $C$ is equal to $\max\{p_e, p_A\}$, where $p_e = \frac{7\lambda + L - \sqrt{\lambda^2 + 14\lambda L + L^2}}{6\lambda}$ and $p_A = 1 - \frac{Ln}{\alpha \lambda^2}$.*

**Proof 12** *We can rewrite $nC$ as follows*

$$
nC = \max\{A(p), B(p)\},
$$

*where $A(p) = \frac{\alpha \lambda^2 p(1-p)}{2n} + \frac{\alpha \lambda^2 (1-p)}{2n} + Lp$ and $B(p) = \frac{\alpha \lambda^2 p(1-p)}{2n} + \frac{\alpha \lambda^2 (1-p)}{2n} + 4\lambda(1-p) - 3\lambda p(1-p)$. The function $B$ is monotonically decreasing as a function of $p$ in $[0,1]$. Note that $B(0) = \frac{\alpha \lambda^2}{2n} + 4\lambda$ and $B(1) = 0$. The function $A$ admits a minimizer equal to $p_A = 1 - \frac{Ln}{\alpha \lambda^2}$. Of course $p_A$ is a probability under the condition that $Ln \leq \alpha \lambda^2$. Thus we consider the following 2 scenarios*

1. *If $Ln > \alpha \lambda^2$ ($p_A < 0$) then $p^* = p_e$*

2. *else $p^* = \max\{p_e, p_A\}$.*

*We conclude in both cases that $p^* = \max\{p_e, p_A\}$.*

## A.2 Addendum to the Experimental Results

**Batch Normalization.** Beside the trainable parameters, the ResNet models contain batch normalization Ioffe & Szegedy (2015) layers that are crucial for training. The logic of batch normalization depends on the estimation of running mean and variance, and these statistics can be pretty personalized for each client in a heterogeneous data regime. The implementation of FedAvg and FedOpt in FedML considers the averaging of these statistics during the aggregation phase. In our implementation, the batch normalization statistics are included into aggregation.

**Step-size.** The step-sizes for FedAvg and FedOpt tuned via selecting step sizes from the following set $\{0.01, 0.1, 0.2, 0.5, 1.0, 2.0, 4.0\}$. We consider the step size for both algorithms to be 0.1. Starting with step size 0.2 algorithms diverge; we also did not use step size schedulers. Additionally, we have tuned number of local epochs for FedAvg from the following set $\{1, 2, 3, 4\}$, and pick 1 local epoch. The batch size is set to 256.

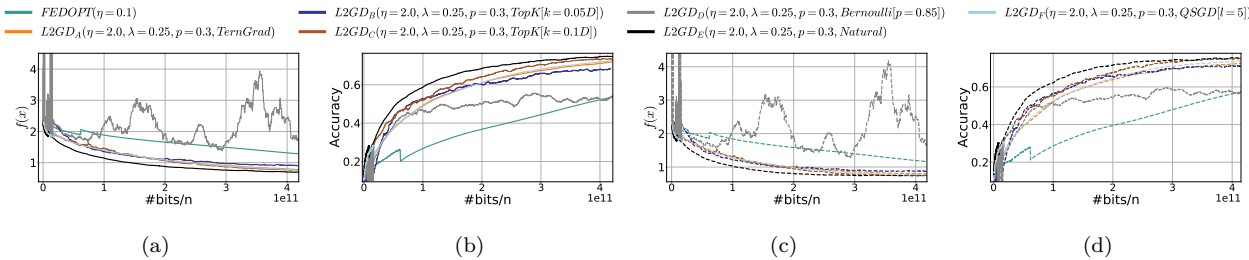

**Figure 10: Training `DenseNet-121` on `CIFAR-10`, with $n = 10$ workers. Loss and Top-1 accuracy on train (a) - (b), and test data (c) - (d).**

**Figure 11: Training `MobileNet` on `CIFAR-10`, with $n = 10$ workers. Loss and Top-1 accuracy on train (a) - (b), and test data (c) - (d).**

**Compressed L2Gd vs. FedOpt.** From the experiments in Section 7.2, Figures 4–6, we realized that FedAvg is not a competitive no-compression baseline for L2GD; see Table 2. FedOpt, on the other hand, remains a competitive no-compression baseline comparable to compressed L2GD. Therefore, we separately measure the performance of compressed L2GD and non-compression FedOpt for training `ResNet-18`, `DenseNet-121`, and `MobileNet`. Figures 9–10 demonstrate that L2GD with natural compressor (that by design has small variance) empirically behaves the best and converges approximately 5 times faster compare to FedOpt. They also show that compressed L2GD with natural compressor sends the least data and drives the loss down the most. At the same time, L2GD with natural compressor reaches the best accuracy for both train and test sets.

**Reproducible research.** See our repository online: `https://github.com/burlachenkok/compressed-fl-l2gd-code`. Our source codes have been constructed on top of the popular federated learning repository: FedML.ai; see `https://github.com/FedML-AI/FedML/commit/3b9b68764d922ce239e0b84aceda986cfa977f96`.

