# OpenReview forum: "Personalized Federated Learning with Communication Compression"
_TMLR — Accepted by TMLR_

### Review · Reviewer_RSnY · 2023-07-08

**Summary Of Contributions:**

This paper developed a new approach for reducing communication costs and perusing personalization in federated learning. However, it missed some important related works and the proof has some flaws.

**Audience:**

Yes

**Claims And Evidence:**

No

**Requested Changes:**

1. The communication-efficient algorithms for federated learning have been extensively studied. However, this paper missed many of them, e.g., [1-4]

2. The proof is not correct. When proving theorem 5, the third step uses Lemma 6, and lemma 6 uses lemma 4. But Lemma 4 uses the strong convexity of the loss function, which does not hold for nonconvex problems.

**Strengths And Weaknesses:**

Strengths

1. The proposed method is well-motivated, and the writing is easy to follow.

2. This paper provides extensive experimental results to demonstrate the performance of the proposed approach.


Weakness

1. The communication-efficient algorithms for federated learning have been extensively studied. However, this paper missed many of them, e.g., [1-4]

2. The proof is not correct. When proving theorem 5, the third step uses Lemma 6, and lemma 6 uses lemma 4. But Lemma 4 uses the strong convexity of the loss function, which does not hold for nonconvex problems.

[1] Qsparse-local-SGD: Distributed SGD with Quantization, Sparsification, and Local Computations

[2] FedPAQ: A Communication-Efficient Federated Learning Method with Periodic Averaging and Quantization

[3] Federated Learning with Compression: Unified Analysis and Sharp Guarantees

[4] On the convergence of communication-efficient local sgd for federated learning

---

> ### Author Response · Authors · 2023-07-11
> **Response to the reviewer's (RSnY) comment**
>
> We would like to sincerely thank the reviewer for their helpful feedback and correctly pointing out the bug in the nonconvex convergence. We thank the reviewers for noting ``The proposed method is well-motivated, and the writing is easy to follow" and also, for mentioning "This paper provides extensive experimental results to demonstrate the performance of the proposed approach." We addressed all the comments, and our revision based on them strengthened the manuscript. Major modifications are indicated in \textcolor{blue}{blue} in the revised version of the manuscript. Below please find our detailed response to all comments.
>
> 1. We have added all the references in the revised version of our manuscript.
> 2. We thank the reviewer for pointing out the issue in nonconvex convergence. Indeed, Lemma 4 uses the strong convexity of the loss function, which does not hold for nonconvex problems. We have fixed this issue by adding a new assumption to bound the stochastic gradient, $G(x^k)$, similar to [1,2]. We changed this accordingly in the revised manuscript.
>
> [1] Sebastian U Stich and Sai Praneeth Karimireddy. The error-feedback framework: Better rates for sgd with
> delayed gradients and compressed updates. The Journal of Machine Learning Research, 21(1):9613–9648,
> 2020.
>
> [2] Atal Sahu, Aritra Dutta, Ahmed M Abdelmoniem, Trambak Banerjee, Marco Canini, and Panos Kalnis.
> Rethinking gradient sparsification as total error minimization. Advances in Neural Information Processing
> Systems, 34:8133–8146, 2021.

---

### Review · Reviewer_HbRJ · 2023-07-13

**Summary Of Contributions:**

The paper follows the “Personalized FL” framework introduced by Hanzely and Richtarik which involves optimizing a regularized cost function that seeks a personalized FL model for every device. In order to solve such a cost function, Hanzely and Richard proposed the loop less local gradient descent (L2GD). However, the L2GD algorithm lacks a compression mechanism between for both master-worker and worker-master communication. The authors propose the compressed L2GD algorithm which utilizes a bidirectional compression technique for the L2GD algorithm. They also provide convergence analysis illustrating their compressed L2GD algorithm has a comparable asymptotic convergence rate as vanilla SGD without compression (in both strongly convex and smooth non-convex case) albeit with an increase in the stochastic gradient variance due to the additional compression. The authors provide empirical studies to illustrate the practicality of their algorithm both within their theoretical framework and beyond (e.g. biased compressors).

**Audience:**

Yes

**Claims And Evidence:**

Yes

**Requested Changes:**

1. How does the optimal rate affect the performance in experiments, i.e. results similar to Figure 3 for the compressed method?

2. If the goal of L2GD is to achieve personalisation, why do the experimental results compare the performance of the global model?


**Strengths And Weaknesses:**

Strengths:

-- The paper is well-written and easy to follow.

-- The Algorithm and the statements of the analysis are clear and interpretable.

--- Extensive Empirical analysis for various compression techniques for L2GD under logistic regression and compressed L2GD on image classifications tasks using DNN.

--- The authors claim that this is the first bidirectional compression technique coupled with a probabilistic communication in the federated learning setting.

Weaknesses:

1. The setting explored in the paper is very specific to the L2GD algorithm.

2. Lack of comparison with other communication efficient FL algorithms. For ex. QUAFL: FEDERATED AVERAGING MADE ASYNCHRONOUS AND COMMUNICATION-EFFICIENT, SignFedAvg: A Unified Stochastic Sign-based Compression for Federated Learning.

3. More details on the compression operators used and their properties could be incorporated to make the paper more complete.

---

> ### Author Response · Authors · 2023-07-30
> **Response to the reviewer's (HbRj) comment**
>
>
> We sincerely thank the reviewer for many positive comments about our work, especially for pointing out that the paper is "well-written and easy to follow" and that "empirical studies to illustrate the practicality of their algorithm both within their theoretical framework and beyond (e.g., biased compressors)." Additionally, the reviewer asked two interesting questions. Below, we answer/comment on each of them. We have furnished them in the revised manuscript.
>
> 1. **How does the optimal rate affect the performance in experiments, i.e., results similar to Figure 3 for the compressed method?**
>
> *Answer:* This is a valid suggestion. As suggested by the reviewer, we perform compressed L2GD to solve Logistic Regression with L2 regularization on $n = 5$ workers, with a1a and a2a datasets. We used natural compressor at the clients and no compression on the master. The results are similar to the no-compression plots (as in Figure 2). Please see the new results in the revised manuscript.
>
> 2. **If the goal of L2GD is to achieve personalization, why do the experimental results compare the performance of the global model?**
>
> *Answer:* In our experiments, we wanted to assess the efficiency of both models --- the local models and the global model (average model), $x_{average} = \frac{1}{n} \sum x_i$. To do this, we used two metrics:
>
> -Loss: we compute the average loss over all the losses of the local models, i.e. $f(x_1,\cdots,x_n) = \frac{1}{n} \sum_i f_i(x_i).$ This allows us to assess the efficiency of the local models.
>
> -Accuracy: we compute the accuracy of the average model, that is, we compute the accuracy using the model $x_{average}.$ As the experiments demonstrate both local models and the average model perform well.
>
> Please note that we performed our experiments on the state-of-the-art FedML benchmarking, and it does not support personalization. Due to this limitation, although we wanted to present the average accuracy over all the accuracies of the local models, we were not able to do so --- Changing the FedML benchmarking for personalization is an involved task. Nevertheless, we compared with compressed communication approaches in the FedML framework that supports a single global model to see how despite personalization, our compressed L2GD performs.

---

### Review · Reviewer_uMx1 · 2023-07-18

**Summary Of Contributions:**

The paper proposes a modification to the Loopless Gradient Descent (L2GD) algorithm of Hanzely and Richtarik [HZ20]. This is an algorithm to solve a variation of federated learning (FL) which allows for some degree of "personalization" and is relevant in settings where there is heterogeneity among the clients' local datasets. The L2GD algorithm essentially switches between (local) gradient update steps and (global) model aggregation steps --- just like regular FL --- but these switches happen probabilistically. It is not too hard to algebraically show that for many model aggregation steps, some amount of communication needs to happen between the central server and the clients.

The algorithmic contribution of the paper is an updated version of this algorithm where both the "uplink" and "downlink" communication is implemented via compression algorithms. This (obviously) reduces communication costs, and the main theoretical contribution of the paper is to show the convergence is not (essentially) affected by such a compression step. Variants of this approach have appeared in the FL literature before; the key novelty seems to be its application to L2GD and therefore personalized FL.


**Audience:**

Yes

**Broader Impact Concerns:**

Not applicable.

**Claims And Evidence:**

No

**Requested Changes:**

*Requested changes*

* Consider pre-emptively clarifying answering some of the questions below in the main text.
* Consider pre-emptively addressing the "incrementality" question up front in the paper.


*Questions*

* How do you initialize $\xi_{-1}$ in Algorithm 1? I understand that for subsequent iterations, this parameter is based on random coin tosses.
* I assume that all the local agents have shared randomness (or at least that the probabilistic bit is broadcast to all agents so that they know what to do in each iteration)? It may be helpful to clarify this in the pseudocode.
* I'm somewhat confused about one point in the algorithm. It seems that $\bar{x}^{k}$ -- the "pristine"/non-compressed average of all client models -- is never made available to all clients, except at initialization. The only information received by the clients are the *noisy* / compressed averages of the model parameters. What if we had a series of coin tosses where $\xi$ was set to 0 (i.e., all the model weights move far away from their original inits), which was then followed by consecutive 1's? In this case, the operation $\bar{x}^{k} = \bar{x}^{k-1}$ would throw an error since $\bar{x}^{k-1}$ is undefined?
I think this discrepancy doesn't affect any convergence analysis (due to Lemma 2) but please check if this intuition is correct.
* It may be helpful to provide a bit more background/context for Assumption 1 -- why this is necessary, whether standard, etc. Also, $C_i$ is at the moment defined as a deterministic compression operator at the start of Section 3, in which case it probably does not make sense to talk about unbiasedness, independence, write down $E_{C_i}$, etc.
* Consider making Fig 2 bigger and more readable. Also, specify the model (logistic regression) in the caption.
* I am not quite sure why "optimal" is in quotes in Section 6 and on Page 12. Consider being more precise.


**Strengths And Weaknesses:**

*Strengths*

* The paper is quite well written (modulo some clarity questions, see below).
* The authors do a good job of surveying the surrounding literature.
* as such the theoretical results (related to the convergence of compressed L2GD) seem sound.
* The experiments make a consistent case that the method works well.

*Weaknesses*

* The proposed algorithm seemed like a fairly incremental change to L2GD, and therefore was not very interesting to me. The main contribution is an concatenation of a known technique (probabilistic compression in both uplink and downlink) with a fairly niche way to do personalized FL (L2GD). I do agree with the authors that personalized FL is an important problem of practical interest, but there are a ton of ways of doing personalization and L2GD does not seem to rank among the methods of choice at the moment. However, I'm not very up tp date with the latest trends in FL, so perhaps there is even now a push towards "exhaustively characterizing" the space of all possible FL approaches, in which case I suppose this paper fills a specific gap in the literature.
* Some clarification questions seem unaddressed; see below for requested changes.

---

> ### Author Response · Authors · 2023-07-30
> **Response to the reviewer's (uMx1) comment**
>
> We would like to sincerely thank the reviewer for providing positive feedback about our work. We thank the reviewer for noting "The authors do a good job of surveying the surrounding literature, as such the theoretical results seem sound." We addressed all the comments and clarified them in the revised version. Our revision based on these comments has strengthened the manuscript. Please find below the answer to each question/comment the reviewer made. We apologize for a minor HTML compiling error in comment 3; we tried our best, but could not resolve it.
>
> 1. **Initialization of $\xi_{-1}$:** The initialization, $\xi_{-1}$ is not important, as it does not impact the Algorithm. In Algorithm 1, we start the for loop (for $k=0,1,…$) by drawing  $\xi_k$. At iteration $k$, we perform an aggregation step if $\xi_k = 1$. That is why to be consistent in the algorithm, we initialized $\xi$ at iteration “-1” by 1 and initialized $\bar{x}_{-1}$ by the average of the initial models, $x_i^0.$ We have added this as a remark in the revised version.
>
> 2. **Shared randomness of the local agents:** Yes, the reviewer is correct. All the agents have access to the same value of $\xi$. We assumed that $\xi$ is drawn at the server side and broadcasted to the agents. We clarified this in the pseudo-code of the revised version of the manuscript.
>
>
> 3. **Comments on the non-compressed average of the client models:** This is an interesting observation, and the reviewer is correct. Please allow us to elaborate further. The only scenario where we use  $\bar{x_k}=\bar{x}_{k-1}$, is the case where for both k-1 and k, we do the averaging instead of performing local steps (the case where we have at least two successive ones for $\xi$). In this case, the mean of all local models is the same at iteration $k-1$ and $k$; please see the bottom of page 5. Therefore, in this scenario, since the master server has already all the local models from the iteration $k-1$, there is no need for the clients to resend their (personalized) models to the master to compute their mean. Instead, the master can compute the mean at iteration $k$ directly from iteration $k-1$ and use it for the averaging. Finally, as the reviewer mentioned, it does not affect the convergence of the compressed L2GD algorithm.
>
> 4. **Comments on Assumption 1:**  We respectfully note that the first two bullet points in Assumption 1 are classic assumptions on compressors and are vastly used in the compressed communication literature for theoretical analysis. The first point assumes the unbiasedness of the compressor, and the second point assumes that the variance of the compressed vector is bounded by the variance of the original vector. Usually, in this context, the compressors are used to compress stochastic quantities (e.g., stochastic gradient for gradient compression), and by satisfying these properties, we get new stochastic quantities with similar properties as those of the original ones (that is, unbiasedness and bounded variance).
>
> The third and fourth points are the independence between the compressors at the participating local agents and the master. This setting is desired in federated learning (FL). Because, in general, the local agents are geographically remote and are free to choose the compressor that is independent of the choice of the other agents. There is no reason to assume any dependence between these quantities since we are in an FL setting where agents are working independently from each other. These assumptions of independence are instrumental for our analysis. We have added this remark in the main paper.
>
> 5. **Bigger Figure 2:** Thank you for pointing out this, we fixed it in the revised version.
>
> 6. **Use of the word *optimal* under quote:** We optimized the complexity bounds of our algorithm as a function of “some hard-to-compute constants in real life” involved in it. Therefore, we put the word optimal in quotes because we are not sure about the exact value of the optimal parameters. However, after the reviewer’s comment, we have omitted the quote.

---

### Author Response · Authors · 2023-08-05
**Summary of Revision**

We thank the reviewers for an overall positive evaluation of our work. We were able to address all the comments and made the "requested changes." Our revision based on them strengthened the manuscript. We indicated the major modifications in **blue** in the revised version of the manuscript. Below we provide a quick summary of the major changes we made.

--- Initialization of $\xi_{-1}$ and shared randomness of the local agents: We clarified them. (Reviewer uMx1)

--- Optimal rates for compressed methods: We performed these sets of experiments and reported them in Figure 3 under *Meta Parameter Study*, see Section 7.1. (Reviewer HbRj)

--- Performance of the global model: We addressed this in depth in Section 7.2: Training DNN models. (Reviewer HbRj)

--- Corrected Proof of Nonconvex convergence: We have fixed this issue by adding a new assumption to bound the stochastic gradient,
$G(x^k)$; see Assumption 3. We changed the proof of Theorem 2 accordingly in the revised manuscript. Our final conclusion remains same: compressed L2GD algorithm has a comparable asymptotic convergence rate as vanilla SGD without compression albeit with an increase in the stochastic gradient variance due to the additional compression. (Reviewer RSnY)

We believe that the review period led to improvements in clarity and experimental validation of the paper. We hope that all of your questions have been answered and are happy engage in further discussions.

---

### Decision · Action_Editors · 2023-09-25

**Recommendation:** Accept as is

**Comment:**

The reviewers' comments have been addressed and all recommended acceptance.

**Audience:**

Yes

**Claims And Evidence:**

Yes